# Skew-Fit: State-Covering Self-Supervised Reinforcement Learning

## Abstract

Autonomous agents that must exhibit flexible and broad capabilities will need to be equipped with large repertoires of skills. Defining each skill with a manually-designed reward function limits this repertoire and imposes a manual engineering burden. Self-supervised agents that set their own goals can automate this process, but designing appropriate goal setting objectives can be difficult, and often involves heuristic design decisions. In this paper, we propose a formal exploration objective for goal-reaching policies that maximizes state coverage. We show that this objective is equivalent to maximizing the entropy of the goal distribution together with goal reaching performance, where goals correspond to full state observations. To instantiate this principle, we present an algorithm called Skew-Fit for learning a maximum-entropy goal distributions. Skew-Fit enables self-supervised agents to autonomously choose and practice reaching diverse goals. We show that, under certain regularity conditions, our method converges to a uniform distribution over the set of valid states, even when we do not know this set beforehand. Our experiments show that it can learn a variety of manipulation tasks from images, including opening a door with a real robot, entirely from scratch and without any manually-designed reward function.

## 1 Introduction

Reinforcement learning (RL) provides an appealing formalism for automated learning of behavioral skills, but separately learning every potentially useful skill becomes prohibitively time consuming, both in terms of the experience required for the agent and the effort required for the user to design reward functions for each behavior. What if we could instead design an unsupervised RL algorithm that automatically explores the environment and iteratively distills this experience into general-purpose policies that can accomplish new user-specified tasks at test time?

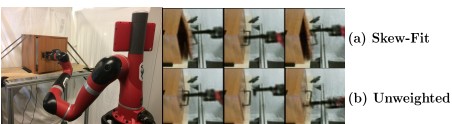

(a) Skew-Fit

(b) Unweighted

Figure 1: Left: Robot learning to open a door with Skew-Fit, without any task reward. Right: Samples from a goal distribution when using (a) Skew-Fit and (b) unweighted (ie. uniform) sampling. When used as goals, the diverse samples from Skew-Fit encourage the robot to practice opening the door more frequently.

For an agent to learn autonomously, it needs an exploration objective. In the absence of any prior knowledge about which states are more useful, an effective exploration scheme is one that visits as many states as possible, allowing a policy to autonomously prepare for user-specified task that it might see at test time. This objective has been formalized as maximizing the entropy of the learned policy's visited state distribution [1] $\mathcal{H}(\mathbf{S})$ (Hazan et al., 2018a), since a policy that maximizes this objective should approach a uniform distribution over valid states. Unfortunately, directly optimizing $\mathcal{H}(\mathbf{S})$ requires an accurate model of the policy and environment (Hazan et al., 2018a). Moreover, even if this optimization were tractable, another short-coming of this objective is that the resulting policy cannot be used to solve new tasks: it only knows how to maximize state entropy. In other words, to develop principled unsupervised RL algorithms that result in useful policies, maximizing $\mathcal{H}(\mathbf{S})$ is not enough. We need a mechanism that allows us to *control* the resulting policy to achieve new tasks at test-time.

---

[1] We consider the distribution over terminal states in a finite horizon task and believe this work can be extended to infinite horizon stationary distributions.

We argue that this can be accomplished by performing *goal-directed exploration*. In addition to maximizing the state entropy, we should be able to control where the policy goes by giving it a goal $\mathbf{G}$ that corresponds to a state that it must reach. Mathematically, a goal-conditioned policy should minimize the conditional entropy over the states given a goal, $\mathcal{H}(\mathbf{S} \mid \mathbf{G})$. This objective provides us with a principled way for training a policy to explore all states, by maximizing $\mathcal{H}(\mathbf{S})$, such that the state that is reached can be controlled by commanding goals, which means minimizing $\mathcal{H}(\mathbf{S} \mid \mathbf{G})$.

Directly using this objective is often intractable, since it requires optimizing the entropy of the marginal state distribution of the policy, $\mathcal{H}(\mathbf{S})$. However, we can sidestep this issue by noting that the objective is the mutual information between the state and the goal, $I(\mathbf{S}; \mathbf{G})$, which can be written as:

$$\mathcal{H}(\mathbf{S}) - \mathcal{H}(\mathbf{S}|\mathbf{G}) = I(\mathbf{S}; \mathbf{G}) = \mathcal{H}(\mathbf{G}) - \mathcal{H}(\mathbf{G}|\mathbf{S}). \tag{1}$$

Equation 1 thus gives an equivalent objective for an unsupervised RL algorithm: the agent should set diverse goals, maximizing $\mathcal{H}(\mathbf{G})$, and learn how to reach them, minimizing $\mathcal{H}(\mathbf{G} \mid \mathbf{S})$.

While the second term is the typical objective studied in goal-conditioned RL (Kaelbling, 1993; Andrychowicz et al., 2017), maximizing the diversity of goals is crucial for effectively learning to reach all possible states. In a new environment, acquiring such a maximum-entropy goal distribution is challenging: how can an agent set diverse goals when it does not even know what states exist?

In this paper, we address this question via a new algorithm, Skew-Fit, which learns to model the uniform distribution over states, given only access to data collected by an autonomous goal-conditioned policy. Our paper makes the following contributions. First, we propose a principled objective for unsupervised RL, based on Equation 1. While a number of prior works ignore the $\mathcal{H}(\mathbf{G})$ term, we argue that jointly optimizing the entire quantity is needed to develop effective and useful exploration. Second, we propose a method called Skew-Fit and prove that, under some regularity conditions, it learns a generative model that converges to a uniform distribution over the goal space, even when the set of valid states is unknown (e.g., as in the case of images). Third, we empirically demonstrate that, when combined with goal-conditioned RL, Skew-Fit allows us to autonomously train goal-conditioned policies that reach diverse states. We test this method on a variety of simulated vision-based robot tasks without any task-specific reward function. In these experiments, Skew-Fit reaches substantially better final performance than prior methods, and learns much more quickly. We also demonstrate that our approach solves a real-world manipulation task, which requires a robot to learn to open a door from scratch in about five hours, directly from images, and without any manually-designed reward function.

## 2 PROBLEM FORMULATION

To ensure that an unsupervised reinforcement learning agent learns to reach all possible states in a controllable way, we maximize the mutual information between the state $\mathbf{S}$ and the goal $\mathbf{G}$, $I(\mathbf{S}; \mathbf{G})$, as stated in Equation 1. This section discusses how to optimize Equation 1 by splitting the optimization into two parts: minimizing $\mathcal{H}(\mathbf{G} \mid \mathbf{S})$ and maximizing $\mathcal{H}(\mathbf{G})$.

### 2.1 MINIMIZING $\mathcal{H}(\mathbf{G} \mid \mathbf{S})$: GOAL-CONDITIONED REINFORCEMENT LEARNING

Standard RL considers a Markov decision process (MDP), which has a state space $\mathcal{S}$, action space $\mathcal{A}$, and unknown dynamics $\rho(\mathbf{s}_{t+1} \mid \mathbf{s}_t, \mathbf{a}_t) : \mathcal{S} \times \mathcal{S} \times \mathcal{A} \mapsto [0, +\infty)$. Goal-conditioned RL also includes a goal space $\mathcal{G}$. For simplicity, we will assume in our derivation that the goal space matches the state space, such that $\mathcal{G} = \mathcal{S}$, though the approach extends trivially to the case where $\mathcal{G}$ is a hand-specified subset of $\mathcal{S}$, such as the global x-y position of a robot. A goal-conditioned policy $\pi(\mathbf{a} \mid \mathbf{s}, \mathbf{g})$ maps a state $\mathbf{s} \in \mathcal{S}$ and goal $\mathbf{g} \in \mathcal{S}$ to a distribution over actions $\mathbf{a} \in \mathcal{A}$, and its objective is to reach the goal, i.e., to make the current state equal to the goal.

Goal-reaching can be formulated as minimizing $\mathcal{H}(\mathbf{G} \mid \mathbf{S})$, and many practical goal-reaching algorithms (Kaelbling, 1993; Lillicrap et al., 2016; Schaul et al., 2015; Andrychowicz et al., 2017; Nair et al., 2018; Pong et al., 2018; Florensa et al., 2018a) can be viewed as approximations to this objective by observing that the optimal goal-conditioned policy will deterministically reach the goal, resulting in a conditional entropy of zero: $\mathcal{H}(\mathbf{G} \mid \mathbf{S}) = 0$. See Appendix E for more details. Our method may thus be used in conjunction with any of these prior goal-conditioned RL methods in order to jointly minimize $\mathcal{H}(\mathbf{G} \mid \mathbf{S})$ and maximize $\mathcal{H}(\mathbf{G})$.

## 2.2 Maximizing $\mathcal{H}(\mathbf{G})$: Setting Diverse Goals

We now turn to the problem of setting diverse goals or, mathematically, maximizing the entropy of the goal distribution $\mathcal{H}(\mathbf{G})$. Let $U_{\mathcal{S}}$ be the uniform distribution over $\mathcal{S}$, where we assume $\mathcal{S}$ has finite volume so that the uniform distribution is well-defined. Let $p_\phi$ be the goal distribution from which goals $\mathbf{G}$ are sampled. Our goal is to maximize the entropy of $p_\phi$, which we write as $\mathcal{H}(\mathbf{G})$. Since the maximum entropy distribution over $\mathcal{S}$ is the uniform distribution $U_{\mathcal{S}}$, maximizing $\mathcal{H}(\mathbf{G})$ may seem as simple as choosing the uniform distribution to be our goal distribution: $p_\phi = U_{\mathcal{S}}$. However, this requires knowing the uniform distribution over valid states, which may be difficult to obtain when $\mathcal{S}$ is a subset of $\mathbb{R}^n$, for some $n$. For example, if the states correspond to images viewed through a robot's camera, $\mathcal{S}$ corresponds to the (unknown) set of valid images of the robot's environment, while $\mathbb{R}^n$ corresponds to all possible arrays of pixel values of a particular size. In such environments, sampling from the uniform distribution $\mathbb{R}^n$ is unlikely to correspond to a valid image of the real world. Sampling uniformly from $\mathcal{S}$ would require knowing the set of all possible valid images, which we assume the agent does not know when starting to explore the environment.

While we cannot sample arbitrary states from $\mathcal{S}$, we can sample states by performing goal-directed exploration. To derive and analyze our method, we introduce a simple model of this process: a goal $\mathbf{G} \sim p_\phi$ is sampled from the goal distribution $p_\phi$, and then the agent attempts to achieve this goal, which results in a distribution of states $\mathbf{S} \in \mathcal{S}$ seen along the trajectory. We abstract this entire process by writing the resulting marginal distribution over $\mathbf{S}$ as $p(\mathbf{S} \mid p_\phi)$. We assume that $p(\mathbf{S} \mid p_\phi)$ has full support, which can be accomplished with an epsilon-greedy goal reaching policy in a communicating MDP. We also assume that the entropy of the resulting state distribution $\mathcal{H}(p(\mathbf{S} \mid p_\phi))$ is no less than the entropy of the goal distribution $\mathcal{H}(p_\phi(\mathbf{S}))$. Without this assumption, a policy could ignore the goal and stay in a single state, no matter how diverse and realistic the goals are. Note that this assumption does **not** require that the entropy of $p(\mathbf{S} \mid p_\phi)$ is strictly larger than the entropy of the goal distribution, $p_\phi$. This simplified model allows us to analyze the behavior of our goal-setting scheme separately from any specific goal-reaching algorithm. We will however show in Section 6 that we can instantiate this approach into a practical algorithm that jointly learns the goal-reaching policy. In summary, our goal is to acquire a maximum-entropy goal distribution $p_\phi$ over valid states $\mathcal{S}$, while only having access to state samples from $p(\mathbf{S} \mid p_\phi)$.

## 3 Skew-Fit: Learning a Maximum Entropy Goal Distribution

Our method, Skew-Fit, learns a maximum entropy goal distribution $p_\phi$ using samples collected from a goal-conditioned policy. We analyze the algorithm and show that Skew-Fit maximizes the entropy of the goal distribution, and present a practical instantiation for unsupervised deep RL.

### 3.1 Skew-Fit Algorithm

To learn a uniform distribution over valid goal states, we present a method that iteratively increases the entropy of a generative model $p_\phi$. In particular, given a generative model $p_{\phi_t}$ at iteration $t$, we would like to train a new generative model $p_{\phi_{t+1}}$ such that $p_{\phi_{t+1}}$ has higher entropy than $p_{\phi_t}$ over the set of valid states. While we do not know the set of valid states $\mathcal{S}$, we can sample states from $p(\mathbf{S} \mid p_{\phi_t})$, resulting in an empirical distribution $p_{\mathrm{emp}_t}$ over the states

$$p_{\mathrm{emp}_t}(\mathbf{s}) \triangleq \frac{1}{N} \sum_{n=1}^{N} \mathbf{1}\{\mathbf{s} = \mathbf{S}_n\}, \quad \mathbf{S}_n \sim p(\mathbf{S} \mid p_{\phi_t}), \tag{2}$$

and use this empirical distribution to train the next generative model $p_{\phi_{t+1}}$. However, if we simply train $p_{\phi_{t+1}}$ to model this empirical distribution, it may not necessarily have higher entropy than $p_{\phi_t}$.

The intuition behind our method is quite simple: rather than fitting a generative model to our empirical distribution, we *skew* the empirical distribution so that rarely visited states are given more weight. See Figure 2 for a visualization of this process. How should we skew the empirical distribution if we want to maximize the entropy of $p_{\phi_{t+1}}$? If we had access to the density of each state, $p_{\mathrm{emp}_t}(\mathbf{S})$, then we could simply weight each state by $1/p_{\mathrm{emp}_t}(\mathbf{S})$. We could then perform maximum likelihood

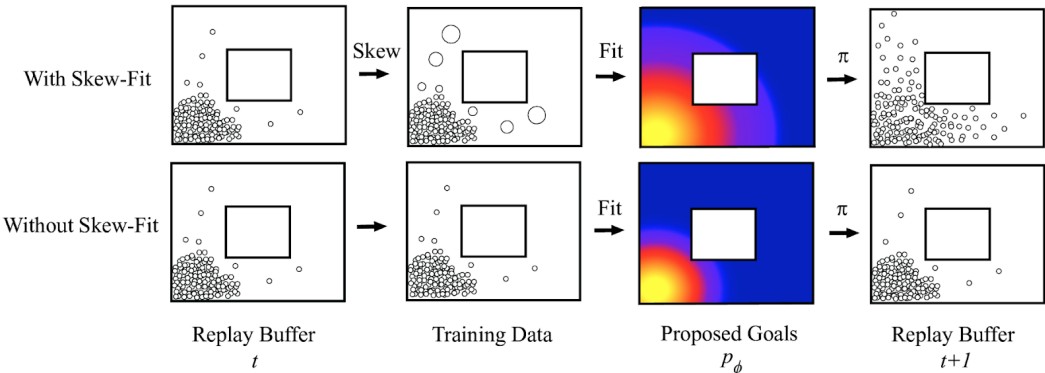

Figure 2: Our method, Skew-Fit, samples goals for goal-conditioned RL in order to induce a uniform state visitation distribution. We start by sampling from our replay buffer, and weighting the states such that rare states are given more weight. We then train a generative model $p_{\phi_{t+1}}$ with the weighted samples. By sampling new states with goals proposed from this new generative model, we obtain a higher entropy distribution of states in our replay buffer at the next iteration.

estimation (MLE) for the uniform distribution by using the following loss to train $\phi_{t+1}$:

$$\mathcal{L}(\phi) = \mathbb{E}_{\mathbf{S} \sim U_{\mathcal{S}}} \left[ \log p_\phi(\mathbf{S}) \right] = \mathbb{E}_{\mathbf{S} \sim p_{\mathrm{emp}_t}} \left[ \frac{U_{\mathcal{S}}(\mathbf{S})}{p_{\mathrm{emp}_t}(\mathbf{S})} \log p_\phi(\mathbf{S}) \right] \propto \mathbb{E}_{\mathbf{S} \sim p_{\mathrm{emp}_t}} \left[ \frac{1}{p_{\mathrm{emp}_t}(\mathbf{S})} \log p_\phi(\mathbf{S}) \right]$$

where we use the fact that the uniform distribution $U_{\mathcal{S}}(\mathbf{S})$ has constant density for all states in $\mathcal{S}$. However, computing this density $p_{\mathrm{emp}_t}(\mathbf{S})$ requires marginalizing out the MDP dynamics, which requires an accurate model of both the dynamics and the goal-conditioned policy.

We avoid needing to model the entire MDP process by approximating $p_{\mathrm{emp}_t}(\mathbf{S})$ with our previous learned generative model: $p_{\mathrm{emp}_t}(\mathbf{S}) \approx p(\mathbf{S} \mid p_{\phi_t}) \approx p_{\phi_t}(\mathbf{S})$. We therefore weight each state by the following weight function

$$w_{t,\alpha}(\mathbf{S}) \triangleq p_{\phi_t}(\mathbf{S})^\alpha, \quad \alpha < 0. \tag{3}$$

where $\alpha$ is a hyperparameter that controls how heavily we weight each state. If our approximation $p_{\phi_t}$ was exact, we could choose $\alpha = -1$ and recover the exact importance sampling procedure described above. If $\alpha = 0$, then this skew step has no effect. By choosing intermediate values of $\alpha$, we can trade off the reliability of our estimate $p_{\phi_t}(\mathbf{S})$ with the speed at which we want to increase the entropy of the goal distribution.

**Variance Reduction**   As described, this procedure relies on importance sampling (IS), which can have high variance, particularly if $p_{\phi_t}(\mathbf{S}) \approx 0$. We therefore choose a class of generative models where the probabilities are prevented from collapsing to zero, as we will describe in Section 4. To further reduce the variance, we train $p_{\phi_{t+1}}$ with sampling importance resampling (SIR) (Rubin, 1988). Rather than sampling from $p_{\mathrm{emp}_t}$ and weighting the update from each sample by $w_{t,\alpha}$, SIR explicitly defines a skewed distribution as

$$p_{\mathrm{skewed}_t}(\mathbf{s}) \triangleq \frac{1}{Z_\alpha} p_{\mathrm{emp}_t}(\mathbf{s}) w_{t,\alpha}(\mathbf{s}), \quad Z_\alpha = \sum_{n=1}^N p_{\mathrm{emp}_t}(\mathbf{S}_n) w_{t,\alpha}(\mathbf{S}_n), \tag{4}$$

where $Z_\alpha$ is the normalizing coefficient and $p_{\mathrm{emp}_t}$ is given by Equation 2. We note that computing $Z_\alpha$ adds little computational overhead, since all of the weights already need to be computed. We then fit the generative model at the next iteration $p_{\phi_{t+1}}$ to $p_{\mathrm{skewed}_t}$ using standard MLE. We found that using SIR resulted in significantly lower variance than IS. See Appendix B.3 for this comparision.

**Goal Sampling Alternative**   Because $p_{\phi_{t+1}} \approx p_{\mathrm{skewed}_t}$, at iteration $t + 1$, one can sample goals from either $p_{\phi_{t+1}}$ or $p_{\mathrm{skewed}_t}$. Sampling goals from $p_{\mathrm{skewed}_t}$ may be preferred if sampling from the learned generative model $p_{\phi_{t+1}}$ is computationally or otherwise challenging. In either case, one still needs to train the generative model $p_{\phi_t}$ to create $p_{\mathrm{skewed}_t}$. In our experiments, we found that both methods perform well.

**Summary**   Overall, Skew-Fit samples data from the environment and weights different samples by their density under the generative model $p_{\phi_t}$. We prove in the next section conditions under which this weighting makes the generative model at the next iteration $p_{\phi_{t+1}}$ have higher entropy. With higher entropy, the $p_{\phi_{t+1}}$ is more likely to generate goals at the frontier of unseen states, which results in more uniform state coverage. Skew-Fit is shown in Figure 2 and summarized in Algorithm 1.

---

**Algorithm 1** Skew-Fit

---

1: **for** Iteration $t = 1, 2, \ldots$ **do**
2:    Collect $N$ states $\{\mathbf{S}_i\}_{i=1}^N$ by sampling goals from $p_{\phi_t}$ (or $p_{\text{skewed}_t}$) and running goal-conditioned policy.
3:    Construct skewed distribution $p_{\text{skewed}_t}$ (Equation 3 and Equation 4).
4:    Fit $p_{\phi_{t+1}}$ to skewed distribution $p_{\text{skewed}_t}$ using MLE.
5: **end for**

---

### 3.2   Skew-Fit Analysis

In this section, we provide conditions under which $p_{\phi_t}$ converges in distribution to the uniform distribution over the state space $\mathcal{S}$. To make this analysis possible, we consider the case where $N \to \infty$, which allows us to study the limit behavior of the goal distribution $p_{\text{skewed}_t}$. Our most general result is stated as follows:

**Lemma 3.1.** *Let $\mathcal{S}$ be a compact set. Define the set of distributions $\mathcal{Q} = \{p : \text{support of } p \text{ is } \mathcal{S}\}$. Let $\mathcal{F} : \mathcal{Q} \mapsto \mathcal{Q}$ be a continuous function and such that $\mathcal{H}(\mathcal{F}(p)) \geq \mathcal{H}(p)$ with equality if and only if $p$ is the uniform probability distribution on $\mathcal{S}$, $U_{\mathcal{S}}$. Define the sequence of distributions $P = (p_1, p_2, \ldots)$ by starting with any $p_1 \in \mathcal{Q}$ and recursively defining $p_{t+1} = \mathcal{F}(p_t)$.*

*The sequence $P$ converges to $U_{\mathcal{S}}$.*

*Proof.* See Appendix Section E.  □

We will apply Lemma 3.1 to be the map from $p_{\text{skewed}_t}$ to $p_{\text{skewed}_{t+1}}$ to show that $p_{\text{skewed}_t}$ converges to $U_{\mathcal{S}}$. If we assume that the goal-conditioned policy and generative model learning procedure are well behaved ( i.e., the maps from $p_{\phi_t}(\mathbf{S})$ to $p_{\text{emp}_t}$ and from $p_{\text{skewed}_t}$ to $p_{\phi_{t+1}}$ are continuous ), then to apply Lemma 3.1, we only need to show that $\mathcal{H}(p_{\text{skewed}_t}) \geq \mathcal{H}(p_{\text{emp}_t})$ with equality if and only if $p_{\text{emp}_t} = U_{\mathcal{S}}$. For the simple case when $p_{\phi_t} = p_{\text{emp}_t}$ identically at each iteration, we prove the convergence of Skew-Fit true for any value of $\alpha \in [-1, 0)$ in Appendix A.3. However, in practice, $p_{\phi_t}$ only approximates $p_{\text{emp}_t}$. To address this more realistic situation, we prove the following result:

**Lemma 3.2.** *Given two distribution $p_{emp_t}$ and $p_{\phi_t}$ where $p_{emp_t} \ll p_{\phi_t}$ [2] and*

$$\text{Cov}_{\mathbf{S} \sim p_{emp_t}} \left[ \log p_{emp_t}(\mathbf{S}), \log p_{\phi_t}(\mathbf{S}) \right] > 0, \tag{5}$$

*define the distribution $p_{skewed_t}$ as in Equation 4. Let $\mathcal{H}_\alpha(\alpha)$ be the entropy of $p_{skewed_t}$ for a fixed $\alpha$. Then there exists a constant $a < 0$ such that for all $\alpha \in [a, 0)$,*

$$\mathcal{H}(p_{skewed_t}) = \mathcal{H}_\alpha(\alpha) > \mathcal{H}(p_{emp_t}).$$

*Proof.* See Appendix Section E.  □

Thus, our generative model $p_{\phi_t}$ does not need to exactly fit the empirical distribution. We merely need for the log densities of $p_{\phi_t}$ and $p_{\text{emp}_t}$ to be correlated, which we expect to happen frequently with an accurate goal-conditioned policy, since $p_{\text{emp}_t}$ is the set of states seen when trying to reach goals from $p_{\phi_t}$. In this case, if we choose negative values of $\alpha$ that are small enough, then the entropy of $p_{\text{skewed}_t}$ will be higher than that of $p_{\text{emp}_t}$. Empirically, we found that $\alpha$ values as low as $\alpha = -1$ performed well.

In summary, we see that under certain assumptions, $p_{\text{skewed}_t}$ converges to $U_{\mathcal{S}}$. Since we train each generative model $p_{\phi_{t+1}}$ by fitting it to $p_{\text{skewed}_t}$, we expect $p_{\phi_t}$ to also converge to $U_{\mathcal{S}}$.

---

[2] $p \ll q$ means that $p$ is absolutely continuous with respect to $q$, i.e. $p(\mathbf{s}) = 0 \implies q(\mathbf{s}) = 0$.

## 4 TRAINING GOAL-CONDITIONED POLICIES WITH SKEW-FIT

Thus far, we have presented and derived Skew-Fit assuming that we have access to a goal-reaching policy, allowing us to separately analyze how we can maximize $\mathcal{H}(\mathbf{G})$. However, in practice we do not have access to such a policy, and in this section we discuss how we concurrently train a goal-reaching policy.

Maximizing $I(\mathbf{S}; \mathbf{G})$ can be done by simultaneously performing Skew-Fit and training a goal conditioned policy to minimize $\mathcal{H}(\mathbf{G} \mid \mathbf{S})$, or, equivalently, maximize $-\mathcal{H}(\mathbf{G} \mid \mathbf{S})$. Maximizing $-\mathcal{H}(\mathbf{G} \mid \mathbf{S})$ requires computing the density $\log p(\mathbf{G} \mid \mathbf{S})$, which may be difficult to compute without strong modeling assumptions. However, for any distribution $q$, the following lower bound for $-\mathcal{H}(\mathbf{G} \mid \mathbf{S})$ holds:

$$-\mathcal{H}(\mathbf{G} \mid \mathbf{S}) = \mathbb{E}_{(\mathbf{G},\mathbf{S}) \sim p_{\phi_t}, \pi} \left[ \log q(\mathbf{G} \mid \mathbf{S}) \right] + D_{\mathrm{KL}}(p \mid q) \geq \mathbb{E}_{(\mathbf{G},\mathbf{S}) \sim p_{\phi_t}, \pi} \left[ \log q(\mathbf{G} \mid \mathbf{S}) \right],$$

where $D_{\mathrm{KL}}$ denotes Kullback–Leibler divergence as discussed by Barber & Agakov (2004). Thus, to minimize $\mathcal{H}(\mathbf{G} \mid \mathbf{S})$, we train a policy to maximize the following reward:

$$r(\mathbf{S}, \mathbf{G}) = \log q(\mathbf{G} \mid \mathbf{S}).$$

For the RL algorithm, we use reinforcement learning with imagined goals (RIG) (Nair et al., 2018), though in principle any goal-conditioned method could be used. RIG is an efficient off-policy goal-conditioned method that solves the vision-based RL problem in a learned latent space. In particular, RIG fits a $\beta$-VAE and uses it to encode all observations and goals into a latent space, which it uses as the state representation. RIG also uses the $\beta$-VAE to compute rewards, $\log q(\mathbf{G} \mid \mathbf{S})$. Unlike RIG, we use the goal distribution from Skew-Fit to sample goals, both for exploration and for relabeling goals during training (Andrychowicz et al., 2017). Since RIG already trains a generative model over states, we reuse this $\beta$-VAE for the generative model $p_\phi$ of Skew-Fit. To make the most use of the data, $p_\phi$ is trained on all visited state rather than only the terminal states, which we found to work well in practice. In other words, our method uses the likelihood estimates from the $\beta$-VAE to choose the probability of sampling each state in Equation 3. To prevent these probabilities from collapsing to zero, we model the posterior of the $\beta$-VAE as a multivariate Gaussian distribution with a fixed variance and only learn the mean. We include a detailed summary of RIG and description our how we combine Skew-Fit and RIG in Appendix C.1.

## 5 RELATED WORK

Many prior methods for training goal-conditioned policies assume that a goal distribution is available to sample from during exploration (Kaelbling, 1993; Schaul et al., 2015; Andrychowicz et al., 2017; Pong et al., 2018). Other methods use data collected from a randomly initialized policy or heuristics based on data collected online to design a non-parametric (Colas et al., 2018b; Warde-Farley et al., 2018; Florensa et al., 2018a; Zhao & Tresp, 2019) or parametric (Péré et al., 2018; Nair et al., 2018) goal distribution. We remark that Warde-Farley et al. (2018) also motivate their work in terms of minimizing a lower bound for $\mathcal{H}(\mathbf{G} \mid \mathbf{S})$. Our work is complementary to these goal-reaching methods: rather than focusing on how to train goal-reaching policies, we propose a principled method for maximizing the entropy of a goal sampling distribution, $\mathcal{H}(\mathbf{G})$.

Our method learns without any task rewards, directly acquiring a policy that can be reused to reach user-specified goals. This stands in contrast to exploration methods that give bonus rewards based on state visitation frequency (Bellemare et al., 2016; Ostrovski et al., 2017; Tang et al., 2017; Savinov et al., 2018; Chentanez et al., 2005; Lopes et al., 2012; Stadie et al., 2016; Pathak et al., 2017; Burda et al., 2018; 2019; Mohamed & Rezende, 2015; Tang et al., 2017; Fu et al., 2017). While these methods can also be used without a task reward, they provide no mechanism for distilling the knowledge gained from visiting diverse states into flexible policies that can be applied to accomplish new goals at test-time: their policies visit novel states, and they quickly forget about them as other states become more novel.

Other prior methods extract reusable skills in the form of latent-variable-conditioned policies, where latent variables can be interpreted as options (Sutton et al., 1999) or abstract skills (Hausman et al., 2018; Gupta et al., 2018b; Eysenbach et al., 2019; Gupta et al., 2018a; Florensa et al., 2017). The

resulting skills may be diverse, but they have no grounded interpretation, while our method can be used immediately after unsupervised training to reach diverse user-specified goals.

Some prior methods propose to choose goals based on heuristics such as learning progress (Baranes & Oudeyer, 2012; Veeriah et al., 2018; Colas et al., 2018a), how off-policy the goal is (Nachum et al., 2018), level of difficulty (Florensa et al., 2018b) or likelihood ranking (Zhao & Tresp, 2019). In contrast, our approach provides a principled framework for optimizing a concrete and well-motivated exploration objective, and can be shown to maximize this objective under regularity assumptions. The work of Hazan et al. (2018b) also provably optimizes a well-motivated exploration objective, but is limited to tabular MDPs, while Skew-Fit is able to handle high dimensional settings such as vision-based continuous control.

## 6 EXPERIMENTS

Our experiments study the following questions: **(1)** Does Skew-Fit empirically result in a goal distribution with increasing entropy? **(2)** In image-based domains, how does Skew-Fit compare to prior work on choosing goals for goal-conditioned RL? **(3)** Can Skew-Fit be applied to a real-world, vision-based robot task?

**Does Skew-Fit Maximize Entropy?** To see the effects of Skew-Fit on goal distribution entropy in isolation of learning a goal-reaching policy, we begin by studying an idealized example where the policy is a near-perfect goal-reaching policy. The MDP is defined on a 2-by-2 unit square-shaped corridor (see Figure 3). At the beginning of an episode, the agent begins in the bottom-left corner and samples a goal from the goal distribution $p_{\phi_t}$. To simulate the stochasticity of the policy and environment, we add a Gaussian noise with standard deviation of $0.05$ to this goal. The policy reaches the state that is closest to this noisy goal and inside the corridor, giving us a state $\mathbf{S}$ to add to our empirical distribution. We compare Skew-Fit to sampling uniformly from the replay buffer (labeled MLE). The $\beta$-VAE hyperparameters used to train $p_{\phi_t}$ are given in Appendix C.5. As seen in Figure 3,

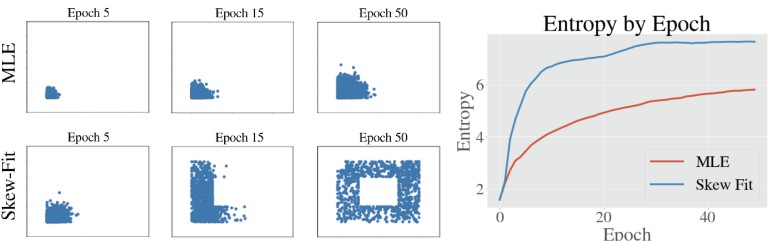

Figure 3: (Left) The set of final states visited by our agent and MLE over the course of training. In contrast to MLE, our method quickly approaches a uniform distribution over the set of valid states. (Right) The entropy of the sample data distribution, which quickly reaches its maximum for Skew-Fit. The entropy was calculated via discretization onto a 60 by 60 grid.

naively using previous experience to set goals results in a policy that primarily sets goal near the initial state distribution and only relies on the stochasticity of the policy and environment to explore. In contrast, Skew-Fit results in quickly learning a high entropy, near-uniform distribution over the state space.

**Vision-Based Continuous Control Tasks** We now evaluate Skew-Fit on a variety of continuous control tasks, where the policy must control a robot arm using only image observations, without access to any ground truth reward signal. We test our method on three different simulated continuous control tasks released by the authors of RIG (Nair et al., 2018): *Visual Door*, *Visual Pusher*, and *Visual Pickup*. To our knowledge, these are the only goal-conditioned, vision-based continuous control environments that are publicly available and used in experimental evaluations in prior work, making them a good point of comparison. See Figure 4 for visuals and Appendix C for details of these environments. The policies are trained in a completely unsupervised manner, without access to any prior information about the state-space or any pre-defined goal-sampling distribution. To evaluate their performance, we sample goal images from a uniform distribution over valid states and report the agent's final distance to the corresponding simulator states (e.g., distance of the object

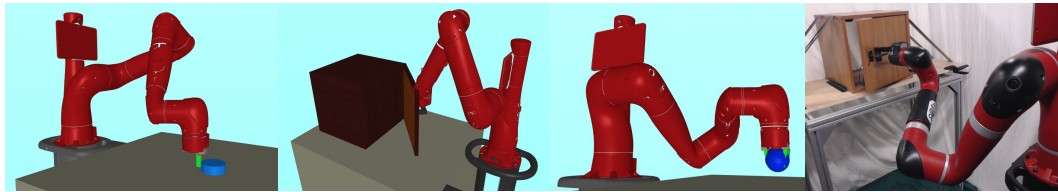

Figure 4: We evaluate on these continuous control environments. From left to right: *Visual Pusher*, a simulated pushing task; *Visual Door*, a door opening task; *Visual Pickup*, a picking task; and *Real World Visual Door*, a real world door opening task. All tasks are solved from images and without any task-specific reward. See Appendix D for details.

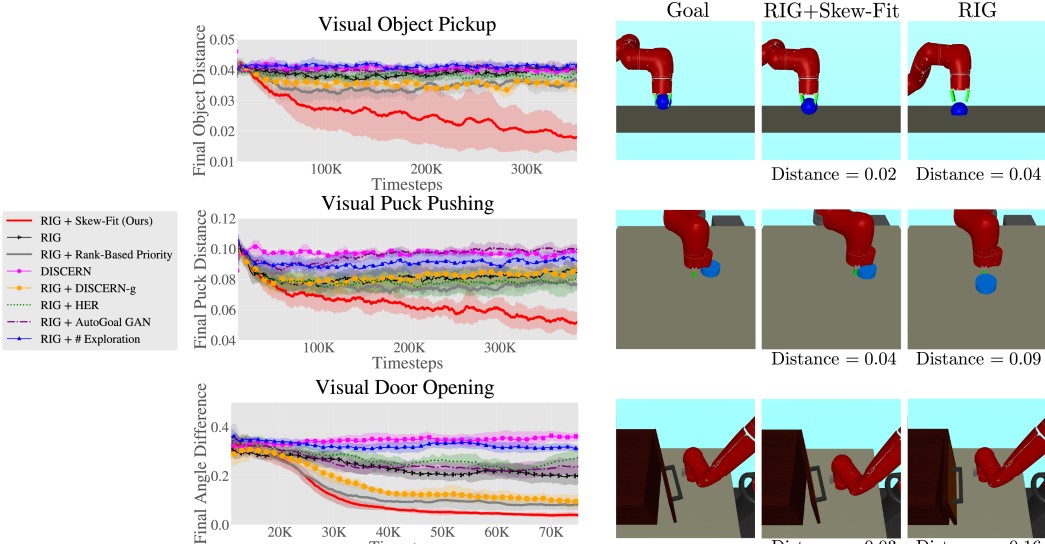

Figure 5: (Left) Learning curves for simulated continuous control experiments. Lower is better. For each environment and method, we show the mean and standard deviation of 6 seeds and smooth temporally across 25 epochs within each seed. Skew-Fit consistently outperforms RIG and various baselines. See the text for description of each method. (Right) The first column displays example test goal images for each environment. In the next two columns, we display final images reached by Skew-Fit and RIG respectively. Under each image is the final distance in state space to provide a notion of the behavior of each method in the plots.

to the target object location), but the agent never has access to this true uniform distribution nor the ground-truth state information during training. While this evaluation method and metric is only practical in simulation, it provides us with a quantitative measure of a policy's ability to reach a broad coverage of goals in a vision-based setting.

We use these domains to compare Skew-Fit to a number of existing methods on goal-sampling. We compare to Warde-Farley et al. (2018), a vision-based method which uses a non-parametric approach based on clustering to sample goals and an image discriminator to compute rewards. We denote this method as **DISCERN**. The other methods that we compare to were developed in non-vision, state-based environments. To ensure a fair comparison across methods, we combine these prior methods with a policy trained using RIG. First, we compare to **RIG** without Skew-Fit. We also compared to RIG using the relabeling scheme described in the hindsight experience replay (labeled **HER**). We compare to curiosity-driven prioritization (**Ranked-Based Priority**) (Zhao & Tresp, 2019), a variant of HER that samples goals for relabeling based on their ranked likelihoods. Florensa et al. (2018b) samples goals from a GAN based on the difficulty of reaching the goal. We compare against this method by replacing $p_\phi$ with the GAN and label it **AutoGoal GAN**. We also separately compare to the goal proposal mechanism proposed by Warde-Farley et al. (2018) and otherwise train the policy with RIG, which we label **DISCERN-g**. Lastly, to demonstrate the difficulty of the exploration challenge in these domains, we compare to **# Exploration** (Tang et al., 2017), an exploration method that assigns bonus rewards based on the novelty of new states. Implementation details of the prior methods is given in Appendix C.3.

We see in Figure 5 that Skew-Fit significantly outperforms prior methods both in terms of task performance and sample complexity. The most common failure mode for prior methods is that the goal distributions collapse, resulting in the agent learning to reach only a fraction of the state space, as shown in Figure 1. For comparison, additional samples of $p_\phi$ when trained with and without Skew-Fit are shown in Appendix B.4. Those images show that without $Skew-Fit$, $p_\phi$ produces a small, non-diverse distribution for each environment: the object is in the same place for pickup, the puck is often in the starting position for pushing, and the door is always closed. In contrast, Skew-Fit proposes goals where the object is in the air and on the ground, where the puck positions are varied, and the door angle changes.

The direct effect of these goal choices can be seen by visualizing more example rollouts for **RIG** and **Skew-Fit**. Due to space constraints, these visuals are in Figure 16 in Appendix B.4. The figure shows that standard RIG only learns to reach states close to the initial position, while Skew-Fit learns to reach the entire state space. A quantitative comparison of the various methods on the pickup task can be seen in Figure 6, which gives the cumulative total exploration pickups for each method. From the graph, we can see that only Skew-Fit learns to pay attention to the object and therefore consistently increases the rate at which the policy picks up the object during exploration. In contrast, the other methods have near constant slopes past 40k steps, meaning that they do not continue to learning, and many methods have a near-constant rate of object lifts throughout all of training.

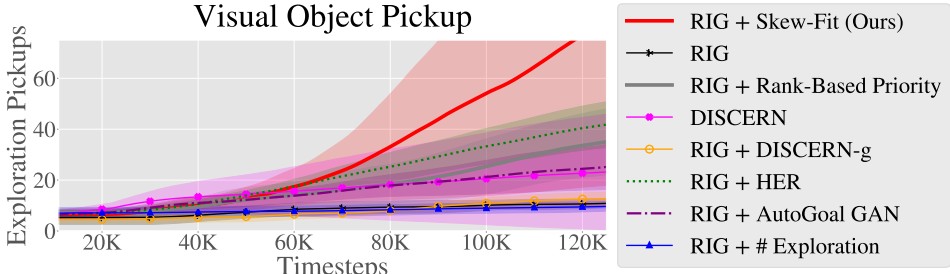

Figure 6: Cumulative total pickups during exploration for each method. The prior methods fail to pay attention to the object and only pick it up at the same rate as the initial policy. In contrast, after seeing the object picked up a few times, Skew-Fit practices picking up the object more often by sampling the appriopriate exploration goals.

**Real-World Vision-Based Robotic Manipulation** We also demonstrate that Skew-Fit scales well to the real world with a door opening task, *Real World Visual Door*. See Figure 4 for a picture of this environment. While a number of prior works have studied RL-based learning of door opening Kalakrishnan et al. (2011); Chebotar et al. (2017), we demonstrate the first method for autonomous learning of door opening without a user-provided, task-specific reward function. As in simulation, we do not provide any goals to the agent and simply let it interact with the door to solve the door opening task from scratch, without any human guidance or reward signal. We train two agents using Skew-Fit with RIG and RIG alone. Unlike in simulation, we cannot measure the difference between the policy's achieved and desired door angle since we do not have access to the true state of the world. Instead, we simply visually denote a binary success/failure for each

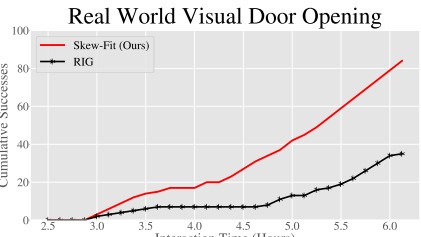

Figure 7: Learning curve for Real World Visual Door environment. We visually label a success if the policy opens the door to the target angle by the last state of the trajectory. Skew-Fit results in considerable sample efficiency gains over prior work on this real-world task.

goal based on whether the last state in the trajectory achieves the target angle. Every seven and a half minutes of interaction time we evaluate on 5 goals and plot the cumulative successes for each method. As Figure 7 shows, standard RIG only starts to open the door after five hours of training. In contrast, Skew-Fit learns to occasionally open the door after three hours of training and achieves a near-perfect success rate after five and a half hours of interaction time, demonstrating that Skew-Fit is a promising technique for solving real world tasks without any human-provided reward function. Videos of Skew-Fit solving this task and the simulated tasks can be viewed on our website.[3]

---

[3] Anonymous while under review: https://sites.google.com/view/skew-fit-iclr-2020

**Additional Experiments** To study the sensitivity of our method to the hyperparameter $\alpha$, we sweep $\alpha$ across the values $[-1, -0.75, -0.5, -0.25, 0]$ on the simulated image-based tasks. Due to space constraints, the sensitivity analysis over the hyperparameter $\alpha$ is in Appendix B, and the results demonstrate that Skew-Fit works across a large range of values for $\alpha$, and $\alpha = -1$ consistently outperform $\alpha = 0$, where the empirical distribution is not skewed. Additionally, Appendix C provides a complete description our method hyper-parameters, including network architecture and RL algorithm hyperparameters.

## 7 CONCLUSION

We presented a formal objective for self-supervised goal-directed exploration, allowing researchers to quantify progress and compare progress when designing algorithms that enable agents to autonomously learn. We also presented Skew-Fit, an algorithm for training a generative model to approximate a uniform distribution over valid states, using data obtained via goal-conditioned reinforcement learning, and our theoretical analysis gives conditions under which Skew-Fit converges to the uniform distribution. When such a model is used to choose goals for exploration and to relabeling goals for training, the resulting method results in much better coverage of the state space, enabling our method to explore effectively. Our experiments show that when we concurrently train a goal-reaching policy using self-generated goals, Skew-Fit produces quantifiable improvements on simulated robotic manipulation tasks, and can be used to learn a door opening skill to reach a $95\%$ success rate directly on a real-world robot, without any human-provided reward supervision.

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

## A   PROOFS

### A.1   PROOF OF LEMMA 3.1

**Lemma A.1.** *Let $\mathcal{S}$ be a compact set. Define the set of distributions $\mathcal{Q} = \{p : \text{ support of } p \text{ is } \mathcal{S}\}$. Let $\mathcal{F} : \mathcal{Q} \mapsto \mathcal{Q}$ be a continuous function and such that $\mathcal{H}(\mathcal{F}(p)) \geq \mathcal{H}(p)$ with equality if and only if $p$ is the uniform probability distribution on $\mathcal{S}$, $U_{\mathcal{S}}$. Define the sequence of distributions $P = (p_1, p_2, \dots)$ by starting with any $p_1 \in \mathcal{Q}$ and recursively defining $p_{t+1} = \mathcal{F}(p_t)$.*

*The sequence $P$ converges to $U_{\mathcal{S}}$.*

*Proof.* The uniform distribution $U_{\mathcal{S}}$ is well defined since $\mathcal{S}$ is compact. Because $\mathcal{S}$ is a compact set, by Prokhorov's Theorem Billingsley (2013), the set $\mathcal{Q}$ is sequentially compact. Thus, $P$ has a convergent subsequence $P' = (p_{k_1}, p_{k_2}, \dots) \subset P$ for $k_1 < k_2 < \dots$ that converges to a distribution $p^* \in \mathcal{Q}$. Because $\mathcal{F}$ is continuous, $p^*$ must be a fixed point of $\mathcal{F}$ since by the convergence mapping theorem, we have that

$$\lim_{i \to \infty} p_{k_i} = p^* \implies \lim_{i \to \infty} \mathcal{F}(p_{k_i}) = \mathcal{H}(p^*)$$

and so

$$
\begin{aligned}
p^* &= \lim_{i \to \infty} p_{k_i} \\
&= \lim_{i \to \infty} \mathcal{F}(p_{k_{i-1}}) \\
&= \mathcal{H}(p^*).
\end{aligned}
$$

The only fixed point of $\mathcal{F}$ is $U_{\mathcal{S}}$ since for any distribution $p$ that is not the uniform distribution, $U_{\mathcal{S}}$, we have that $\mathcal{H}(\mathcal{F}(p)) > \mathcal{H}(p)$ which implies that $\mathcal{F}(p) \neq p$. Thus, $P'$ converges to the only fixed point, $U_{\mathcal{S}}$. Since the entropy cannot decrease, then entropy of the distributions in $P$ must also converge to the entropy of $U_{\mathcal{S}}$. Lastly, since entropy is a continuous function of distribution, $P$ must converge to $U_{\mathcal{S}}$. $\qquad\square$

### A.2   PROOF OF LEMMA 3.2

**Lemma A.2.** *Given two distribution $p(x)$ and $q(x)$ where $p \ll q$ and*

$$0 < \text{Cov}_p[\log p(X), \log q(X)] \tag{6}$$

*define the distribution $p_\alpha$ as*

$$p_\alpha(x) = \frac{1}{Z_\alpha} p(x) q(x)^\alpha$$

*where $\alpha \in \mathbb{R}$ and $Z_\alpha$ is the normalizing factor. Let $\mathcal{H}_\alpha(\alpha)$ be the entropy of $p_\alpha$. Then there exists a constant $a > 0$ such that for all $\alpha \in [-a, 0)$,*

$$\mathcal{H}_\alpha(\alpha) > \mathcal{H}_\alpha(0) = \mathcal{H}(p). \tag{7}$$

*Proof.* Observe that $\{p_\alpha : \alpha \in [-1, 0]\}$ is a one-dimensional exponential family

$$p_\alpha(x) = e^{\alpha T(x) - A(\alpha) + k(x)}$$

with log carrier density $k(x) = \log p(x)$, natural parameter $\alpha$, sufficient statistic $T(x) = \log q(x)$, and log-normalizer $A(\alpha) = \int_{\mathcal{X}} e^{\alpha T(x) + k(x)} dx$. As shown in Nielsen & Nock (2010), the entropy of a distribution from a one-dimensional exponential family with parameter $\alpha$ is given by:

$$\mathcal{H}_\alpha(\alpha) \triangleq \mathcal{H}(p_\alpha) = A(\alpha) - \alpha A'(\alpha) - \mathbb{E}_{p_\alpha}[k(X)]$$

The derivative with respect to $\alpha$ is then

$$
\begin{aligned}
\frac{d}{d\alpha} \mathcal{H}_\alpha(\alpha) &= -\alpha A''(\alpha) - \frac{d}{d\alpha} \mathbb{E}_{p_\alpha}[k(x)] \\
&= -\alpha A''(\alpha) - \mathbb{E}_\alpha[k(x)(T(x) - A'(\alpha)] \\
&= -\alpha \text{Var}_{p_\alpha}[T(x)] - \text{Cov}_{p_\alpha}[k(x), T(x)]
\end{aligned}
$$

where we use the fact that the $n$th derivative of $A(\alpha)$ give the $n$ central moment, i.e. $A'(\alpha) = \mathbb{E}_{p_\alpha}[T(x)]$ and $A''(\alpha) = \mathrm{Var}_{p_\alpha}[T(x)]$. The derivative of $\alpha = 0$ is

$$\frac{d}{d\alpha}\mathcal{H}_\alpha(0) = -\mathrm{Cov}_{p_0}[k(x), T(x)]$$
$$= -\mathrm{Cov}_p[\log p(x), \log q(x)]$$

which is negative by assumption. Because the derivative at $\alpha = 0$ is negative, then there exists a constant $a > 0$ such that for all $\alpha \in [-a, 0]$, $\mathcal{H}_\alpha(\alpha) > \mathcal{H}_\alpha(0) = \mathcal{H}(p)$. $\qquad\square$

### A.3 SIMPLE CASE PROOF

We prove the convergence directly for the (even more) simplified case when $p_\theta = p(\mathbf{S} \mid p_{\phi_t})$ using a similar technique:

**Lemma A.3.** *Assume the set $\mathcal{S}$ has finite volume so that its uniform distribution $U_\mathcal{S}$ is well defined and has finite entropy. Given any distribution $p(\mathbf{s})$ whose support is $\mathcal{S}$, recursively define $p_t$ with $p_1 = p$ and*

$$p_{t+1}(\mathbf{s}) = \frac{1}{Z_\alpha^t}p_t(\mathbf{s})^\alpha, \quad \forall \mathbf{s} \in \mathcal{S}$$

*where $Z_\alpha^t$ is the normalizing constant and $\alpha \in [0, 1)$.*

*The sequence $(p_1, p_2, \dots)$ converges to $U_\mathcal{S}$, the uniform distribution $\mathcal{S}$.*

*Proof.* If $\alpha = 0$, then $p_2$ (and all subsequent distributions) will clearly be the uniform distribution. We now study the case where $\alpha \in (0, 1)$.

At each iteration $t$, define the one-dimensional exponential family $\{p_\theta^t : \theta \in [0, 1]\}$ where $p_\theta^t$ is

$$p_\theta^t(\mathbf{s}) = e^{\theta T(\mathbf{s}) - A(\theta) + k(\mathbf{s})}$$

with log carrier density $k(\mathbf{s}) = 0$, natural parameter $\theta$, sufficient statistic $T(\mathbf{s}) = \log p_t(\mathbf{s})$, and log-normalizer $A(\theta) = \int_\mathcal{S} e^{\theta T(\mathbf{s})} d\mathbf{s}$. As shown in Nielsen & Nock (2010), the entropy of a distribution from a one-dimensional exponential family with parameter $\theta$ is given by:

$$\mathcal{H}_\theta^t(\theta) \triangleq \mathcal{H}(p_\theta^t) = A(\theta) - \theta A'(\theta)$$

The derivative with respect to $\theta$ is then

$$\frac{d}{d\theta}d\mathcal{H}_\theta^t(\theta) = -\theta A''(\theta)$$
$$= -\theta\mathrm{Var}_{\mathbf{s}\sim p_\theta^t}[T(\mathbf{s})]$$
$$= -\theta\mathrm{Var}_{\mathbf{s}\sim p_\theta^t}[\log p_t(\mathbf{s})] \qquad (8)$$
$$\leq 0$$

where we use the fact that the $n$th derivative of $A(\theta)$ is the $n$ central moment, i.e. $A''(\theta) = \mathrm{Var}_{\mathbf{s}\sim p_\theta^t}[T(\mathbf{s})]$. Since variance is always non-negative, this means the entropy is monotonically decreasing with $\theta$. Note that $p_{t+1}$ is a member of this exponential family, with parameter $\theta = \alpha \in (0, 1)$. So

$$\mathcal{H}(p_{t+1}) = \mathcal{H}_\theta^t(\alpha) \geq \mathcal{H}_\theta^t(1) = \mathcal{H}(p_t)$$

which implies

$$\mathcal{H}(p_1) \leq \mathcal{H}(p_2) \leq \dots.$$

This monotonically increasing sequence is upper bounded by the entropy of the uniform distribution, and so this sequence must converge.

The sequence can only converge if $\frac{d}{d\theta}\mathcal{H}_\theta^t(\theta)$ converges to zero. However, because $\alpha$ is bounded away from 0, Equation 8 states that this can only happen if

$$\mathrm{Var}_{\mathbf{s}\sim p_\theta^t}[\log p_t(\mathbf{s})] \to 0. \qquad (9)$$

Because $p_t$ has full support, then so does $p_\theta^t$. Thus, Equation 9 is only true if $\log p_t(\mathbf{s})$ converges to a constant, i.e. $p_t$ converges to the uniform distribution. $\qquad\square$

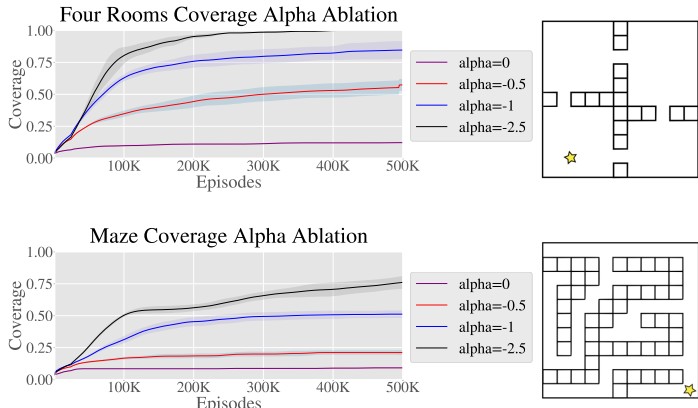

Figure 8: (Top) Coverage over time on the classic 4-room domain, shown on the right. (Bottom) Coverage over time on a more challenging maze domain, shown on the right. In both cases, we see that not using Skew-Fit ($\alpha = 0$) results in significantly slower learning that primarily stays near the start (yellow star).

# B    ADDITIONAL EXPERIMENTS

## B.1    SKEW-FIT FOR EXPLORING LOW-DIMENSIONAL SPACES

Skew-Fit is a general method that enables exploration when it is infeasible to sample goal states uniformly across the entire state space. While the experiments in Section 6 focused on image-based state spaces, there exists many low-dimensional domains in which we know that the goal space is a subset of $\mathbb{R}^d$ for some $d < n$, but the exact goal space is still unknown. This scenario is quite common in domains such as robotics: we know that we want an agent to move the position of its center of mass (CoM), but we do not know the set of valid CoM positions, as this requires knowing the geometry of all potential obstacles a priori. We conduct a series of experiments that study whether Skew-Fit enable effectively exploration in these state spaces containing unknown obstacles.

**2D Maze Navigation with Oracle Policy**    To study the impact of Skew-Fit on exploration in isolation of learning a goal-reaching policy, our first set of experiments use a near-perfect policy that reaches the goal state and then takes a step in a random direction (while taking wall-collisions into account). The random step size is Gaussian with a standard deviation of $0.1$ units, and the size of each square shown in Figure 8 is $1.8$ units. Due to the relatively small step size, the agent cannot rely on random actions to explore the environment and must instead learn to set goals that are progressively farther and farther from the initial state. The first environment is the Four Rooms environment (Sutton et al., 1999), shown in Figure 8 (top). This environment requires a policy to explore four different rooms, each of which requires passing through a narrow doorway. The maze environment (Figure 8, bottom) presents a more challenging exploration problem and consists of various long corridors that require setting goals progressively deeper into the maze. In both domains, setting goals near the state state (represented by the yellow star) and taking small actions will result in minimal exploration. To measure exploration, we discretize the space into squares (see Figure 8 for square sizes) and measure what fraction of the squares the agent has ever visited during exploration. We see in Figure 8 that using Skew-Fit significantly improves exploration, whereas training $p_\phi$ on samples drawn uniformly from the replay buffer ($\alpha = 0$) results in little exploration.

**2D Navigation with Learned Policy**    Next, we reproduce the 2D navigation environment experiment from Section 6, and replace the oracle goal-reacher with a goal-reaching policy that is simultaneously trained with the goal setter. The policy outputs velocities with maximum speed of one. Evaluation goals are chosen uniformly over the valid states. The hyperparameters for this experiment are given in Table 2. In Figure 9a, we can see that a policy trained with a goal distribution trained by Skew-Fit consistently learns to reach all goals, whereas a goal distribution trained with uniform sampling, labeled MLE, results in a policy that fails to reach states far from the starting position (the bottom left corner).

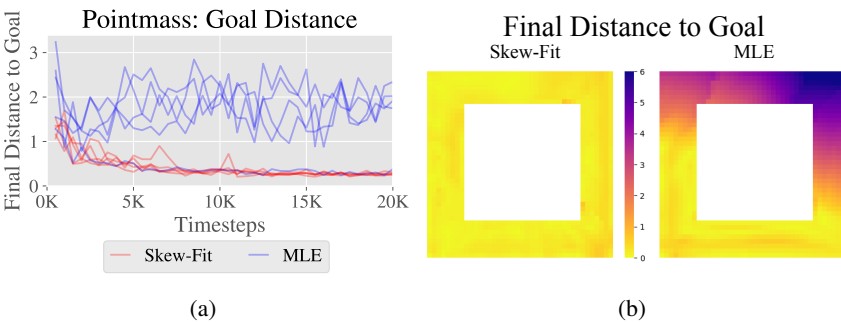

(a)             (b)

Figure 9: (a) Comparison of Skew-Fit vs MLE goal sampling on final distance to goal on RL version of the pointmass environment. Skew-Fit consistently learns to solve the task, while MLE often fails. (b) Heatmaps of final distance to each possible goal location for Skew-Fit and MLE. Skew-Fit learns a good policy over the entire state space, but MLE performs poorly for states far away from the starting position (the bottom left corner).

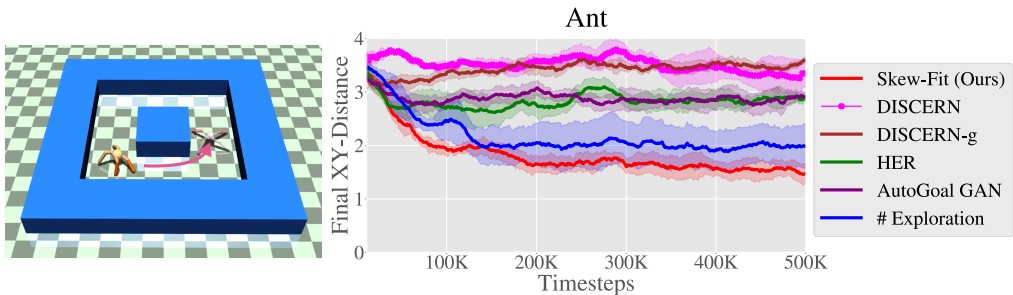

Figure 10: (Left) Ant navigation environment. (Right) Evaluation on reaching joint and XY position. Policies are trained from state. Reward is L2-norm between the current and target joint angle and XY position concatenated together. We use Skew-Fit to sample goals for relabeling and exploration, and compare to other goal sampling methods. See main paper for description of baselines.

**Quadruped "Ant" Locomotion with Learned Policy**      Lastly, we test Skew-Fit in an exploration task that requires training a simulated quadruped "ant" robot to navigate to random XY positions in a plane, as shown in Figure 10. The input to the policy is the joint and velocity of each angle and the reward is the distance to the goal XY-position. While the goal space is known to reside in the XY-plane, the agent does not know about the location of the center obstacle, and so it must still learn about the set of valid goals by controlling its 8 joint actuators. More details of the environment are in Appendix D. We see in Figure 10 that Skew-Fit outperforms prior methods both in terms of learning speed and final performance, demonstrating that Skew-Fit accelerates exploration in non-vision domains that contains unknown goal spaces.

## B.2 SENSITIVITY ANALYSIS

**Sensitivity to RL Algorithm**      In our experiments, we combined Skew-Fit with soft actor critic (SAC) (Haarnoja et al., 2018). We conduct a set of experiments to test whether Skew-Fit may be used with other RL algorithms for training the goal-conditioned policy. To that end, we replaced SAC with twin delayed deep deterministic policy gradient (TD3) (Fujimoto et al., 2018) and ran the same Skew-Fit experiments on Visual Door, Visual Pusher, and Visual Pickup. In Figure 11, we see that Skew-Fit performs consistently well with both SAC and TD3, demonstrating that Skew-Fit is beneficial across multiple RL algorithms.

**Sensitivity to $\alpha$ Hyperparameter**      We study the sensitivity of the $\alpha$ hyperparameter by testing values of $\alpha \in [-1, -0.75, -0.5, -0.25, 0]$ on the Visual Door and Visual Pusher task. The results are included in Figure 12 and shows that our method is robust to different parameters of $\alpha$, particularly for the more challenging Visual Pusher task. Also, the method consistently outperform $\alpha = 0$, which is equivalent to sampling uniformly from the replay buffer.

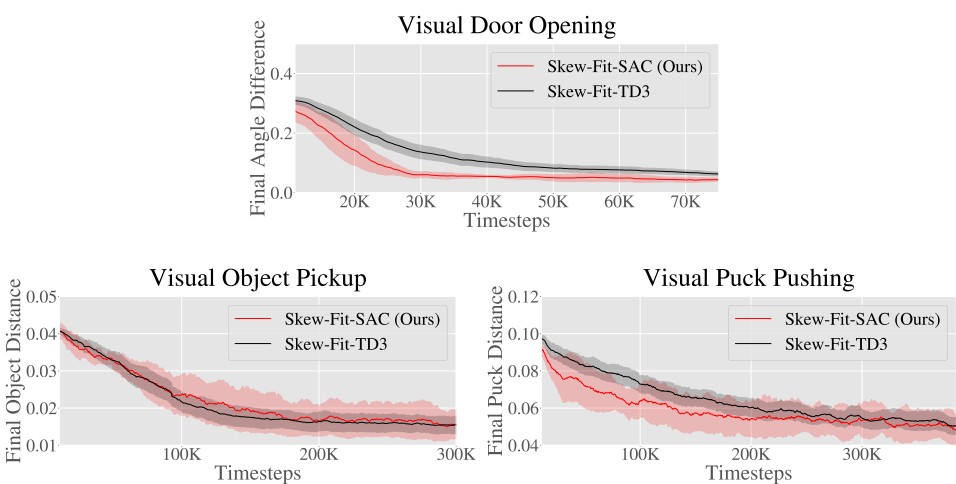

Figure 11: We compare using SAC (Haarnoja et al., 2018) and TD3 (Fujimoto et al., 2018) as the underlying RL algorithm on Visual Door, Visual Pusher and Visual Pickup. We see that Skew-Fit works consistently well with both SAC and TD3, demonstrating that Skew-Fit may be used with various RL algorithms.

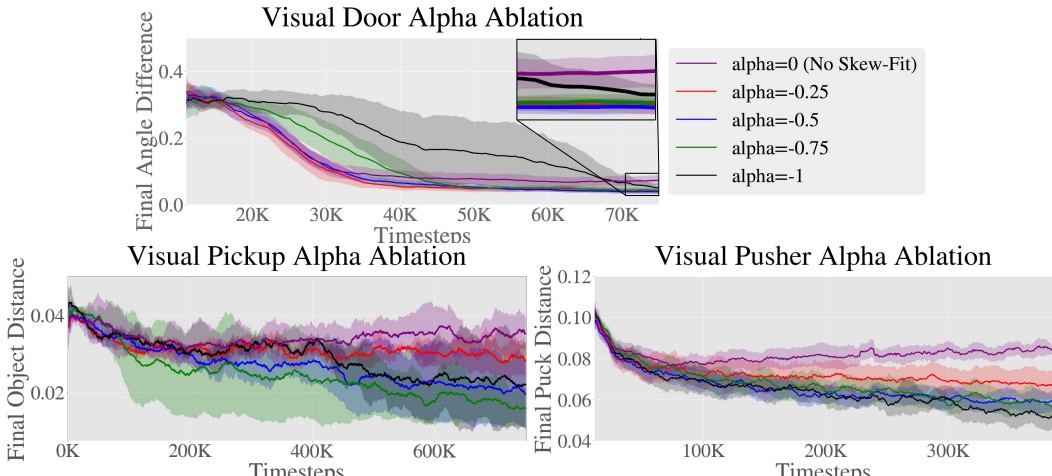

Figure 12: We sweep different values of $\alpha$ on Visual Door, Visual Pusher and Visual Pickup. Skew-Fit helps the final performance on the Visual Door task, and outperforms No Skew-Fit (alpha=0) as seen in the zoomed in version of the plot. In the more challenging Visual Pusher task, we see that Skew-Fit consistently helps and halves the final distance. Similarly, in we observe that Skew-Fit consistently outperforms No Skew-fit on Visual Pickup. Note that alpha=-1 is not always the optimal setting for each environment, but performs strongly in each case in terms of final performance.

| Method | NLL |
|---|---|
| MLE on uniform (oracle) | 20175.4 |
| Skew-Fit on unbalanced | 20175.9 |
| MLE on unbalanced | 20178.03 |

Table 1: Despite training on a unbalanced Visual Door dataset (see Figure 7 of paper), the negative log-likelihood (NLL) of Skew-Fit evaluated on a uniform dataset matches that of a VAE trained on a uniform dataset.

### B.3 Variance Ablation

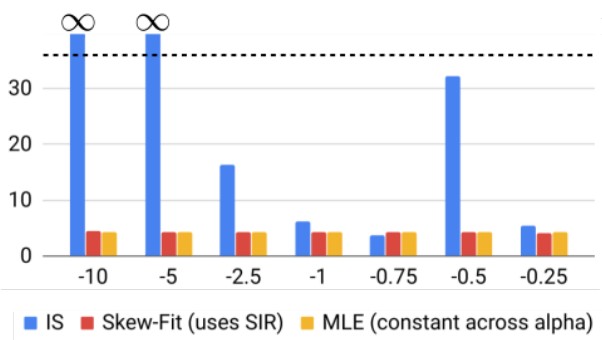

Figure 13: Gradient variance averaged across parameters in last epoch of training VAEs. Values of $\alpha$ less than $-1$ are numerically unstable for importance sampling (IS), but not for Skew-Fit.

We measure the gradient variance of training a VAE on an unbalanced Visual Door image dataset with Skew-Fit vs Skew-Fit with importance sampling (IS) vs no Skew-Fit (labeled MLE). We construct the imbalanced dataset by rolling out a random policy in the environment and collecting the visual observations. Most of the images contained the door in a closed position; in a few, the door was opened. In Figure 13, we see that the gradient variance for Skew-Fit with IS is catastrophically large for large values of $\alpha$. In contrast, for Skew-Fit with SIR, which is what we use in practice, the variance is relatively similar to that of MLE. Additionally we trained three VAE's, one with MLE on a uniform dataset of valid door opening images, one with Skew-Fit on the unbalanced dataset from above, and one with MLE on the same unbalanced dataset. As expected, the VAE that has access to the uniform dataset gets the lowest negative log likelihood score. This is the oracle method, since in practice we would only have access to imbalanced data. As shown in Table 1, Skew-Fit considerably outperforms MLE, getting a much closer to oracle log likelihood score.

### B.4 Goal and Performance Visualization

We visualize the goals sampled from Skew-Fit as well as those sampled when using the prior method, RIG (Nair et al., 2018). As shown in Figure 14 and Figure 15, the generative model $p_\phi$ results in much more diverse samples when trained with Skew-Fit. We we see in Figure 16, this results in a policy that more consistently reaches the goal image.

## C Implementation Details

### C.1 RIG with Skew-Fit Summary

Algorithm 2 provides detailed pseudo-code for how we combined our method with RIG. Steps that were removed from the base RIG algorithm are highlighted in blue and steps that were added are highlighted in red. The main differences between the two are (1) sampling exploration goals from the buffer using $p_{skewed}$ instead of the VAE prior, (2) relabeling with replay buffer goals sampled using $p_{skewed}$ instead of from the VAE prior, and (3) training the VAE on replay buffer data data sampled using $p_{skewed}$ instead of uniformly.

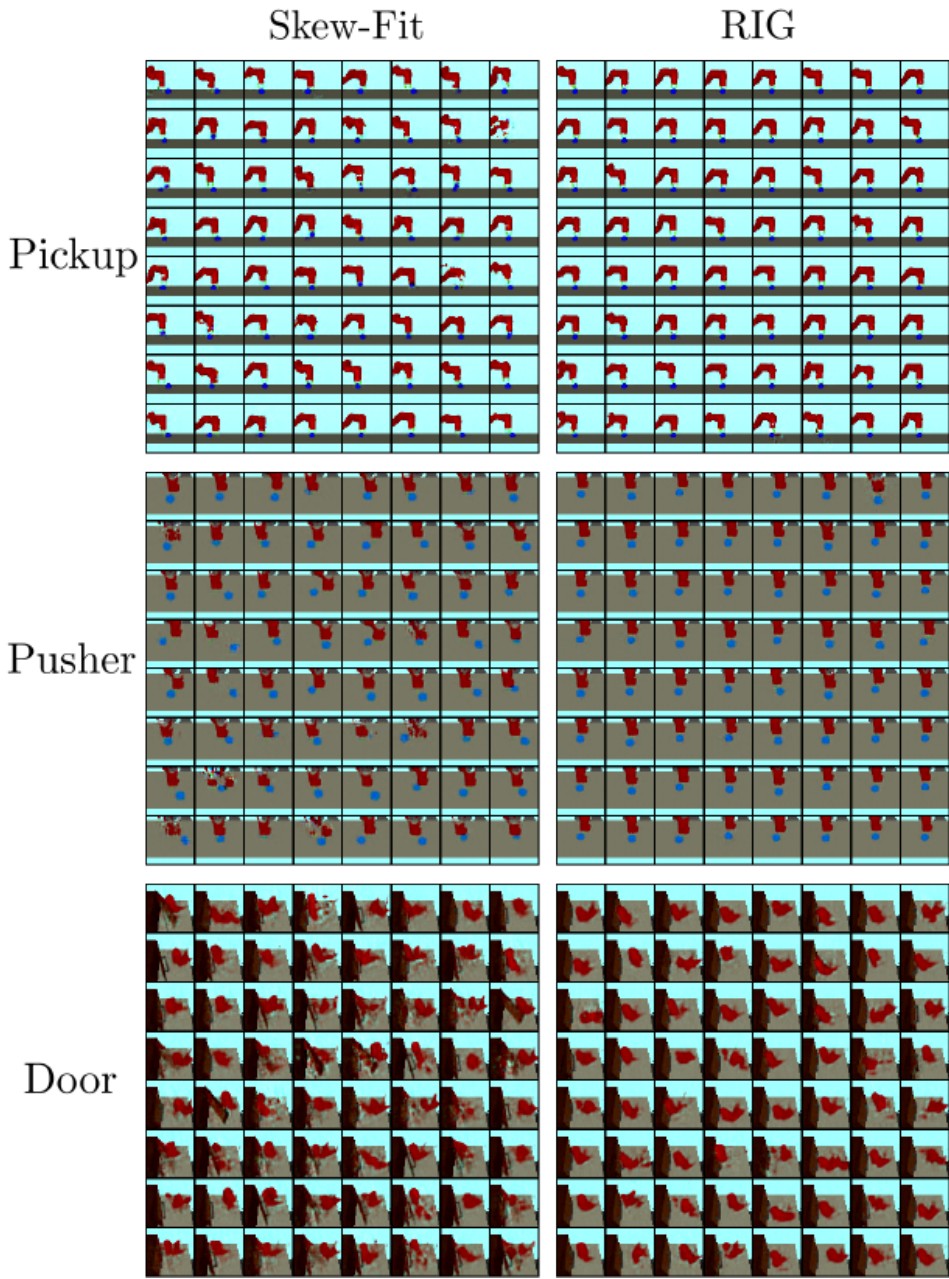

Figure 14: Proposed goals from the VAE for RIG and with Skew-Fit on the *Visual Pickup*, *Visual Pusher*, and *Visual Door* environments. Standard RIG produces goals where the door is closed and the object and puck is in the same position, while RIG + Skew-Fit proposes goals with varied puck positions, occasional object goals in the air, and both open and closed door angles.

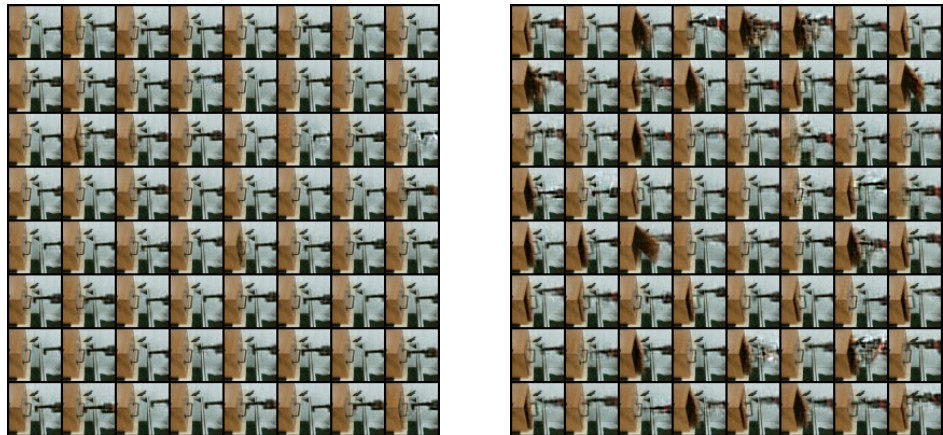

Figure 15: Proposed goals from the VAE for RIG (left) and with RIG + Skew-Fit (right) on the *Real World Visual Door* environment. Standard RIG produces goals where the door is closed while RIG + Skew-Fit proposes goals with both open and closed door angles.

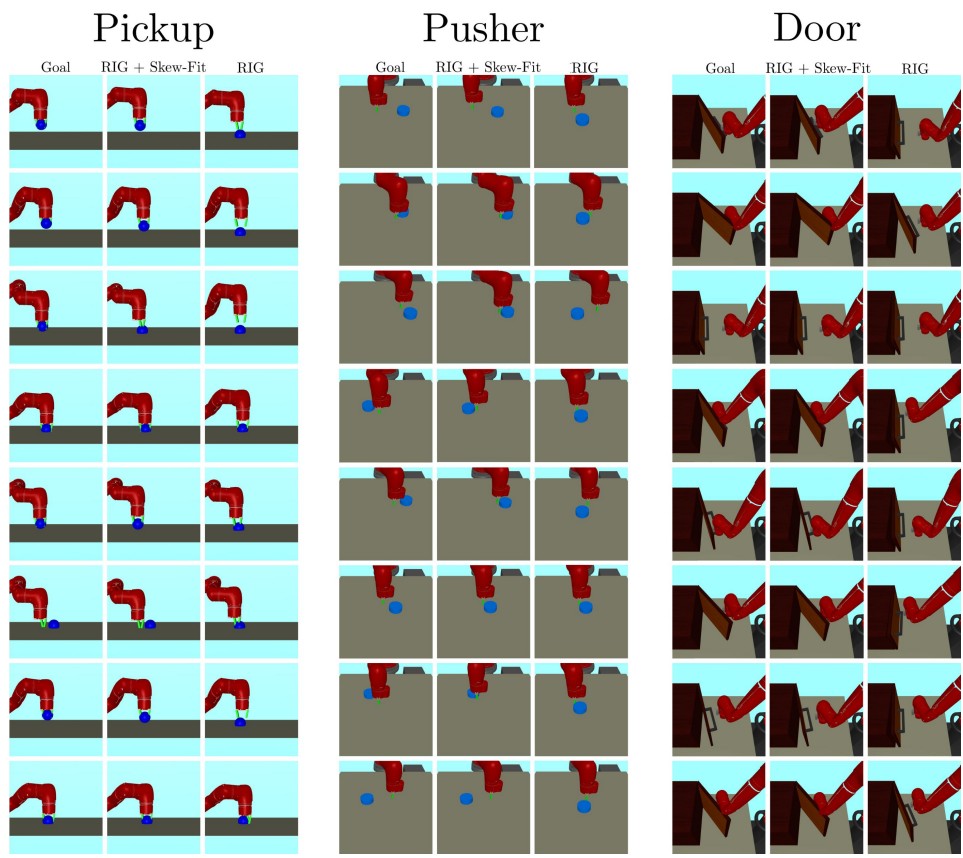

Figure 16: Example reached goals by Skew-Fit and RIG. The first column of each environment section specifies the target goal while the second and third columns show reached goals by Skew-Fit and RIG. Both methods learn how to reach goals close to the initial position, but only Skew-Fit learns to reach the more difficult goals.

## C.2 LIKELIHOOD ESTIMATION USING $\beta$-VAE

We estimate the density under the VAE by using a sample-wise approximation to the marginal over $x$ estimated using importance sampling:

$$p_{\phi_t}(x) = \mathbb{E}_{z \sim q_{\theta_t}(z|x)} \left[ \frac{p(z)}{q_{\theta_t}(z|x)} p_{\psi_t}(x \mid z) \right]$$

$$\approx \frac{1}{N} \sum_{i=1}^{N} \left[ \frac{p(z)}{q_{\theta_t}(z|x)} p_{\psi_t}(x \mid z) \right].$$

where $q_\theta$ is the encoder, $p_\psi$ is the decoder, and $p(z)$ is the prior, which in this case is unit Gaussian. We found that sampling $N = 10$ latents for estimating the density worked well in practice.

## C.3 IMPLEMENTATION OF PRIOR WORK

We replaced TD3 (Fujimoto et al., 2018) with soft actor critic (SAC) from Haarnoja et al. (2018) for all the methods that use RIG, including Skew-Fit.. This is in contrast to the original RIG Nair et al. (2018) paper which used TD3 Fujimoto et al. (2018). We found that maximum entropy policies in general improved the performance of RIG, and that we did not need to add noise on top of the stochastic policy's noise. For our RL network architectures and training scheme, we use fully connected networks for the policy, Q-function and value networks with two hidden layers of size 400 and 300 each. We also delay training any of these networks for 10000 time steps in order to collect sufficient data for the replay buffer as well as to ensure the latent space of the VAE is relatively stable (since we train the VAE online in this setting). As in RIG, we train a goal-conditioned value functions Schaul et al. (2015) using hindsight experience replay Andrychowicz et al. (2017), relabelling $50\%$ of exploration goals as goals sampled from the VAE prior $\mathcal{N}(0, 1)$ and $30\%$ from future goals in the trajectory. In the prior RIG method, the VAE was pre-trained on a uniform sampling of images from the state space of each environment. In order to ensure a fair comparison to Skew-Fit, we forego pre-training and instead train the VAE alongside RL, using the variant described in the RIG paper.

## C.4 VISION-BASED CONTINUOUS CONTROL EXPERIMENTS

In our experiments, we use an image size of 48x48. For our VAE architecture, we use a modified version of the architecture used in the original RIG paper Nair et al. (2018). Our VAE has three convolutional layers with kernel sizes: 5x5, 3x3, and 3x3, number of output filters: 16, 32, and 64 and strides: 3, 2, and 2. We then have a fully connected layer with the latent dimension number of units, and then reverse the architecture with de-convolution layers. We vary the latent dimension of the VAE, the $\beta$ term of the VAE and the $\alpha$ term for Skew-Fit based on the environment. Additionally, we vary the training schedule of the VAE based on the environment. See the table at the end of the appendix for more details. Our VAE has a Gaussian decoder with identity variance, meaning that we train the decoder with a mean-squared error loss.

When training the VAE alongside RL, we found the following two schedules to be effective for different environments:

1. For first $5K$ steps: Train VAE using standard MLE training every 500 time steps for 1000 batches. After that, train VAE using Skew-Fit every 500 time steps for 200 batches.
2. For first $5K$ steps: Train VAE using standard MLE training every 500 time steps for 1000 batches. For the next $45K$ steps, train VAE using Skew-Fit every 500 steps for 200 batches. After that, train VAE using Skew-Fit every 1000 time steps for 200 batches.

We found that initially training the VAE without Skew-Fit improved the stability of the algorithm. This is due to the fact that density estimates under the VAE are constantly changing and inaccurate during the early phases of training. Therefore, it made little sense to use those estimates to prioritize goals early on in training. Instead, we simply train using MLE training for the first $5K$ timesteps, and after that we perform Skew-Fit according to the VAE schedules above. Table 3 lists the hyper-parameters that were shared across the continuous control experiments. Table 4 lists hyper-parameters specific to each environment. Additionally, Appendix C.1 shows the combined RIG + Skew-Fit algorithm.

| Hyper-parameter | Value |
|---|---|
| Algorithm | TD3 Fujimoto et al. (2018)[a] |
| # training batches per time step | 1 |
| Q network hidden sizes | $400, 300$ |
| Policy network hidden sizes | $400, 300$ |
| Q network and policy activation | ReLU |
| Exploration Noise | None |
| RL Batch Size | 1024 |
| Discount Factor | 0.99 |
| Path length | 25 |
| Reward Scaling | 100 |
| Number of steps per epoch | 5000 |

Table 2: Hyper-parameters used for 2D RL experiment (Figure 9a).

[a]We expect similar performance had we used SAC.

| Hyper-parameter | Value | Comments |
|---|---|---|
| # training batches per time step | 2 | Marginal improvements after 2 |
| Exploration Noise | None (SAC policy is stochastic) | Did not tune |
| RL Batch Size | 1024 | smaller batch sizes work as well |
| VAE Batch Size | 64 | Did not tune |
| Discount Factor | 0.99 | Did not tune |
| Reward Scaling | 1 | Did not tune |
| Path length | 100 | Did not tune |
| Replay Buffer Size | 100000 | Did not tune |
| Number of Latents for Estimating Density ($N$) | 10 | Marginal improvements beyond 10 |

Table 3: General hyper-parameters used for all continuous control experiments.

| Hyper-parameter | Visual Pusher | Visual Door | Visual Pickup | Real World Visual Door |
|---|---|---|---|---|
| Path Length | 50 | 100 | 50 | 100 |
| $\beta$ for $\beta$-VAE | 20 | 20 | 30 | 60 |
| Latent Dimension Size | 4 | 16 | 16 | 16 |
| $\alpha$ for Skew-Fit | $-1$ | $-1/2$ | $-1$ | $-1/2$ |
| VAE Training Schedule | 2 | 1 | 2 | 1 |
| Sample Goals From | $p_\phi$ | $p_{\text{skewed}}$ | $p_{\text{skewed}}$ | $p_{\text{skewed}}$ |

Table 4: Environment specific hyper-parameters

---

**Algorithm 2** RIG and RIG + Skew-Fit. Blue text denotes RIG specific steps and red text denotes RIG + Skew-Fit specific steps

**Require:** VAE encoder $q_\phi$, VAE decoder $p_\psi$, policy $\pi_\theta$, goal-conditioned value function $Q_w$, $\alpha$, VAE Training Schedule.

1: Collect $\mathcal{D} = \{s^{(i)}\}$ using exploration policy.
2: Train $\beta$-VAE on data uniformly sampled from $\mathcal{D}$.
3: Fit prior $p(z)$ to latent encodings $\{\mu_\phi(s^{(i)})\}$.
4: **for** $n = 0, ..., N - 1$ episodes **do**
5:    Sample latent goal from prior $z_g \sim p(z)$.
6:    Sample latent goal $e(s')$ from $(s, a, s', z_g) \sim \mathcal{R}$ using $p_\phi$ if $\mathcal{R}$ not empty. Otherwise, use $z_g \sim p(z)$.
7:    Sample initial state $s_0 \sim E$.
8:    **for** $t = 0, ..., H - 1$ steps **do**
9:      Get action $a_t \sim \pi_\theta(e(s_t), z_g)$.
10:     Get next state $s_{t+1} \sim p(\cdot \mid s_t, a_t)$.
11:     Store $(s_t, a_t, s_{t+1}, z_g)$ into replay buffer $\mathcal{R}$.
12:     Sample transition $(s, a, s', z_g) \sim \mathcal{R}$.
13:     Encode $z = e(s), z' = e(s')$.
14:     (Probability 0.5) replace $z_g$ with $z'_g \sim p(z)$.
15:     (Probability 0.5) replace $z_g$ with $e(s')$ where $(s, a, s', z_g) \sim R$ using $p_\phi$
16:     Compute new reward $r = -||z' - z_g||$.
17:     Minimize Bellman Error using $(z, a, z', z_g, r)$.
18:    **end for**
19:    **for** $t = 0, ..., H - 1$ steps **do**
20:      **for** $i = 0, ..., k - 1$ steps **do**
21:       Sample future state $s_{h_i}, t < h_i \leq H - 1$.
22:       Store $(s_t, a_t, s_{t+1}, e(s_{h_i}))$ into $\mathcal{R}$.
23:      **end for**
24:    **end for**
25:    Construct skewed replay buffer distribution $p_\phi$ using data from $\mathcal{R}$ with Equation 4
26:    **if** $total\_steps < 5000$ **then**
27:     Fine-tune $\beta$-VAE on data uniformly sampled from $\mathcal{R}$ according to VAE Training Schedule.
28:    **else**
29:     Fine-tune $\beta$-VAE on data uniformly sampled from $\mathcal{R}$ according to VAE Training Schedule.
30:     Fine-tune $\beta$-VAE on data sampled from $\mathcal{R}$ using $p_\phi$ according to VAE Training Schedule.
31:    **end if**
32: **end for**

---

### C.5 ORACLE 2D NAVIGATION EXPERIMENTS

We initialize the VAE to the middle of the environment for *Maze*, and the bottom left corner of the environment for *Four Rooms*. Both the encoder and decoder have 2 hidden layers with [400, 300] units, ReLU hidden activations, and no output activations. The VAE has a latent dimension of 8 and a Gaussian decoder trained with mean-squared error loss, batch size of 256, and 1000 batches at each iteration. The VAE is trained on the exploration data buffer every 1000 rollouts.

## D ENVIRONMENT DETAILS

*Point-Mass*: In this environment, an agent must learn to navigate a square-shaped corridor (see Figure 3). The observation is the 2D position, and the agent must specify a velocity as the 2D action. The reward at each time step is the negative distance between the achieved position and desired position.

*Maze*: A 20 x 20 2D pointmass environment in the shape of a maze. The observation is the 2D position of the agent, and the agent must specify a target 2D position as the action. The dynamics of the environment are the following: first, the agent is teleported to the target position, specified by the action. Then a gaussian change in position with mean $0$ and standard deviation $0.1$ is then applied. If the action would result in the agent moving through or into a wall, then the agent will be stopped at the wall instead.

*Four Rooms*: A 20 x 20 2D pointmass environment in the shape of four rooms (Sutton et al., 1999). The observation space, actions space, and environment dynamics are the same as the *Maze* environment above.

*Ant*: A MuJoCo ant environment with the same corridor as the *Point-Mass* environment. The observation is a 2D position, orientation, joint angles, and velocity of the joint angles of the ant. The observation space is 29 dimensions. The agent controls the ant through the joints, which is 8 dimensions. The goal is a target 2D position, and the reward is the negative Euclidean distance between the achieved 2D position and target 2D position.

*Visual Pusher*: A MuJoCo environment with a 7-DoF Sawyer arm and a small puck on a table that the arm must push to a target position. The agent controls the arm by commanding $x, y$ position for the end effector (EE). The underlying state is the EE position, $e$ and puck position $p$. The evaluation metric is the distance between the goal and final puck positions. The hand goal/state space is a 10x10

cm$^2$ box and the puck goal/state space is a 30x20 cm$^2$ box. Both the hand and puck spaces are centered around the origin. The action space ranges in the interval $[-1, 1]$ in the x and y dimensions.

*Visual Door*: A MuJoCo environment with a 7-DoF Sawyer arm and a door on a table that the arm must pull open to a target angle. Control is the same as in *Visual Pusher*. The evaluation metric is the distance between the goal and final door angle, measured in radians. In this environment, we do not reset the position of the hand or door at the end of each trajectory. The state/goal space is a 5x20x15 cm$^3$ box in the $x, y, z$ dimension respectively for the arm and an angle between $[0, .83]$ radians. The action space ranges in the interval $[-1, 1]$ in the x, y and z dimensions.

*Visual Pickup*: A MuJoCo environment with the same robot as *Visual Pusher*, but now with a different object. The object is cube-shaped, but a larger intangible sphere is overlaid on top so that it is easier for the agent to see. Moreover, the robot is constrained to move in 2 dimension: it only controls the $y, z$ arm positions. The $x$ position of both the arm and the object is fixed. The evaluation metric is the distance between the goal and final object position. For the purpose of evaluation, $75\%$ of the goals have the object in the air and $25\%$ have the object on the ground. The state/goal space for both the object and the arm is 10cm in the $y$ dimension and 13cm in the $z$ dimension. The action space ranges in the interval $[-1, 1]$ in the $y$ and $z$ dimensions.

*Real World Visual Door*: A Rethink Sawyer Robot with a door on a table. The arm must pull the door open to a target angle. The agent controls the arm by commanding the $x, y, z$ velocity of the EE. Our controller commands actions at a rate of up to 10Hz with the scale of actions ranging up to 1cm in magnitude. The underlying state and goal is the same as in *Visual Door*. Again we do not reset the position of the hand or door at the end of each trajectory. We obtain images using a Kinect Sensor. The state/goal space for the environment is a 10x10x10 cm$^3$ box. The action space ranges in the interval $[-1, 1]$ (in cm) in the x, y and z dimensions. The door angle lies in the range $[0, 45]$ degrees.

## E GOAL-CONDITIONED REINFORCEMENT LEARNING MINIMIZES $\mathcal{H}(\mathbf{G} \mid \mathbf{S})$

Some goal-conditioned RL methods such as Warde-Farley et al. (2018); Nair et al. (2018) present methods for minimizing a lower bound for $\mathcal{H}(\mathbf{G} \mid \mathbf{S})$, by approximating $\log p(\mathbf{G} \mid \mathbf{S})$ and using it as the reward. Other goal-conditioned RL methods (Kaelbling, 1993; Lillicrap et al., 2016; Schaul et al., 2015; Andrychowicz et al., 2017; Pong et al., 2018; Florensa et al., 2018a) are not developed with the intention of minimizing the conditional entropy $\mathcal{H}(\mathbf{G} \mid \mathbf{S})$. Nevertheless, one can see that goal-conditioned RL generally minimizes $\mathcal{H}(\mathbf{G} \mid \mathbf{S})$ by noting that the optimal goal-conditioned policy will deterministically reach the goal. The corresponding conditional entropy of the goal given the state, $\mathcal{H}(\mathbf{G} \mid \mathbf{S})$, would be zero, since given the current state, there would be no uncertainty over the goal (the goal must have been the current state since the policy is optimal). So, the objective of goal-conditioned RL can be interpreted as finding a policy such that $\mathcal{H}(\mathbf{G} \mid \mathbf{S}) = 0$. Since zero is the minimum value of $\mathcal{H}(\mathbf{G} \mid \mathbf{S})$, then goal-conditioned RL can be interpreted as minimizing $\mathcal{H}(\mathbf{G} \mid \mathbf{S})$.

