# OpenReview forum: "Skew-Fit: State-Covering Self-Supervised Reinforcement Learning"
_ICLR.cc/2020/Conference — Reject_

### Official Review · AnonReviewer1 · 2019-10-19
**Official Blind Review #1**

**Rating:** 3

**Review:**

Summary :

The paper proposes an exploratory objective that can maximize state coverage in RL. They show that a formal objective for maximizing state coverage is equivalent to maximizing the entropy of a goal distribution. The core idea is to propose a method to maximize entropy of a goal distribution, or a state distribution since goals are full states. They show that the proposed method to maximize the state or goal distribution can lead to diverse exploration behaviour sufficient for solving complex image based manipulation tasks.


Comments and Questions :

	- The core idea is to maximize the entropy of the state visitation frequency H(s). It is not clear from the paper whether the authors talk about the normalized discounted weighting of states (a distribution) or the stationary distribution? The entropy of the state visitation distribution only deals with valid states - but I am not sure what it means to maximize the entropy of this term exactly in terms of exploration, since it is neither the discounted weighting of states or the stationary distribution for an infinite horizon task?
	- The authors do mention that maximizing the entropy of H(s) is not sufficient - so instead suggests for maxmizing entropy of H(s|g). But why is this even sufficient for exploration - if I do not consider new tasks at test time but only the training task? How is this a sufficient exploration objective? Furthermore, since it is the conditional entropy given goal states, the fundamental idea of this is not clear from the paper.
	- Overall, I am not convinced that an objective based on H(s|g) is equivalent to an maximizing H(s), and why is this even a good objective for exploration? The meaning of H(s) to me is a bit vague from the text (due to reasons above) and therefore H(s|g) does not convince to be a good exploration objective either?
	- The paper then talks about the MI(S;G) to be maximized for exploration - what does this MI formally mean? I understand the breakdown from equation 1, but why is this a sufficient exploration objective? There are multiple ideas introduced at the same time - the MI(s;g) and talking about test time and training time exploration - but the idea itself is not convincing for a sufficient exploration objective. In light of this, I am not sure whether the core idea of the paper is convincing enough to me.
	- I think the paper needs more theoretical insights and details to show why this form of objective based on the MI(s;g) is good enough for exploration. Theoretically, there are a lot of details missing from the paper, and the paper simply proposes the idea of MI(s;g) and talks about formal or computationally tractable ways of computing this term. While the proposed solutuon to compute MI(s;g) seems reasonable, I don't think there is enough contribution or details as to why is maximizing H(s) good for exploration in the first place.
	- Experimentally, few tasks are proposed comparing skew-fit with other baselines like HER and AutoGoal GAN - but the differences in all the results seem negligible (example : Figure 5).
	- I am not sure why the discussion of goal conditioned policies is introduced rightaway. To me, a more convincing approach would have been to first discuss why H(s) and the entropy of this is good for exploration (discounted weighting or stationary state distribution and considering episodic and  infinite horizon tasks). If H(s) is indeed a difficult or not sufficient term to maximize the entropy for, then it might make sense to introduce goal conditioned policies? Following then, it might be convincing to discuss why goal conditioned policies are indeed required, and then tractable ways of computing MI(s;g).
	- Experimentally, I think the paper needs significantly more work - especially considering hard exploration tasks (it might be simple setups too like mazes to begin with), and then to propose a set of new experimental results, without jumping directly to image based tasks as discussed here and then comparing to all the goal conditioned policy baselines.

Overall, I would recommend to reject this paper, as I am not convinced by the proposed solution, and there are lot of theoretical details missing from the paper. It skips a lot of theoretical insights required to propose a new exploration based objective, and the paper proposes a very specific solution for a set a very specific set of experimental setups.





**Experience Assessment:**

I have published one or two papers in this area.

**Review Assessment: Checking Correctness Of Derivations And Theory:**

I carefully checked the derivations and theory.

**Review Assessment: Checking Correctness Of Experiments:**

I carefully checked the experiments.

**Review Assessment: Thoroughness In Paper Reading:**

I read the paper thoroughly.

---

> ### Author Response · Authors · 2019-11-07
> **Re: Official Blind Review #1**
>
> Thank you for the suggestion and detailed review. As suggested, we have modified the introduction to expand our discussion around H(s). We also answer the questions about the use of H(s) and H(s|g) below, and describe how the experimental results do in fact show that Skew-Fit substantially outperforms prior methods. We believe that these clarifications address the major criticisms raised in your review, but we would be happy to address any other points or discuss this further.
>
> Q: Why should H(s) used as an exploration objective?
>
> The goal of our method is to learn a policy that can reach any possible goal state, in the absence of a single user-specified task reward. Prior work has already argued that the entropy of the state distribution H(s) is a suitable exploration objective [1]. In our case, p(s) represents the distribution over terminal states in a finite horizon task, though we believe extensions to infinite horizon stationary distributions should also be possible. Unfortunately, maximizing H(s) by itself does not necessarily provide for a useful policy in the absence of a user-specified reward. For example, if we maximize state coverage by using reward bonuses based on state novelty [2,3,4,5], then, in the absence of user-specified rewards, the resulting policy will only reach the latest states deemed novel.  We instead would like for this policy to be reusable, by, e.g., being able to control what state it reaches. This observation motivates the inclusion of the second term -- H(s|g) -- which amounts to training the policy to effectively (with high probability) reach the commanded goal, while being able to visit as many states/goals as possible. Our overall objective is therefore to maximize H(s) - H(s | g), since maximizing only H(s) does not result in a useful policy. As you pointed out, this has the added benefit that the corresponding algorithm is tractable by using Equation 1, whereas directly maximizing H(s) is difficult. We have accordingly modified the introduction to (1) discuss prior work, (2) raise the concern with directly maximizing H(s), and (3) include a more specific definition of H(s).
>
> Q: What’s the intuition behind the new objective MI(S;G)?
>
> The mutual information provides an equivalent interpretation of our new objective: the new objective changes the exploration objective from “uniformly visit all the states,” as prior work has advocated, to a two stage process: first uniformly set goals over the state space (maximize H(g)) and then separately learn to reach those goals (minimize H(g | s)). At the optimum, the exploration policy will uniformly visits all states, and has the added benefit that we obtain a goal-conditioned policy that can be reused to reach goals.
>
>
> Experiments
> We understand that there were concerns over the significance of the results. We find this concern surprising, as there is a clear difference between Skew-Fit and the next best prior work in Figure 5. Specifically, for the pickup task, Skew-Fit is the only method that makes significant progress: no prior method consistently picks up the object (Figure 6), and Skew-Fit’s final distance is approximately half that of the next best method. For the pushing tasks, the next best method results in a final distance that is 1.5 times worse than that of Skew-Fit, with an average score that is 3-4 standard deviations away from the average score of Skew-Fit. On the door task, some prior methods perform only slightly worse than Skew-Fit. However, we note that this task is much easier than the other tasks (the x-axis more than 4x shorter than the other tasks), as prior work [6] using these environments has also observed. Lastly, the difference on the real-robot experiments are particularly pronounced, with a final success rate double that of the prior method. While we acknowledge that the presentation of the results in the plots could be improved, the results themselves show that Skew-Fit is substantially better than all prior methods that we compared with.
>
> We agree that it is informative to include a simplified experiment that does not directly jump to using goal-conditioned policies nor images. Therefore, Figure 3 of Section 6 analyzes a simplified 2D navigation task. While we did not have room to include in the main paper, Figure 9 of the appendix provides an “in between” experiment that does not contain images, but does include goal-conditioned policies.

---

> > ### Author Response · Authors · 2019-11-07
> > **Citations for: Re: Official Blind Review #1**
> >
> > Due to space constraints, the citations are included in this separate comment:
> >
> > [1] Hazan, Elad, et al. "Provably Efficient Maximum Entropy Exploration." International Conference on Machine Learning. 2019.
> > [2] Bellemare, M, et al.. Unifying count-based exploration and intrinsic motivation. NeurIPS. 2016.
> > [3] Tang, H., et al. #Exploration: A Study of Count-Based Exploration for Deep Reinforcement Learning. NeurIPS, 2017.
> > [4] Burda, Y., et. al. Large-scale study of curiosity-driven learning. ICLR. 2019.
> > [5] Burda, Y., et. al. "Exploration by random network distillation." ICLR. 2019.
> > [6] Nair, Ashvin, et al. “Visual Reinforcement Learning with Imagined Goals. Neural Information Processing Systems. 2019.

---

> > > ### Comment · AnonReviewer1 · 2019-11-13
> > > **Clarifications, further comments and doubts on technical novelty. It seems like the contribution is somewhat marginal. Not fully convinced about the usefulness of this approach.**
> > >
> > > Thank you for your detailed response. It is certainly very helpful. However, I have few other comments and questions :
> > >
> > > - It seems that this approach is useful in the case where there are no user-specified rewards (my understanding is you mean a sparse reward setting). However, as you mention, that p(s) in your case represents the distribution over terminal states for a finite horizing setting - I find it difficult to understand what this distribution actually means in an episodic setting.
> > >
> > > What do you exactly mean by a distribution over terminal states? My understanding is this would be the normalized occupancy measure, where the occupancy is only over the terminal states? Given this, why should we even consider this type of p(s) and why not the normalized discounted weighting of states - and maximize this distribution accordingly? I do agree that simply maximizing this might not be sufficient, which is why you include the H(s|g) term - but that raises further concerns in my opinion, as follows :
> > >
> > > With the current formulation, we would now require distributing goals across the state space - this is similar to Hindsight Experience Replay and related approaches? How are these goals determined in the first place - is it more like a random sampling of states that you consider as goals - and then encouraging the agent to reach these goals?
> > >
> > > Isn't this method then a very slight modification of HER and other related papers? To me, it seems like that makes the contribution very marginal - since instead of HER, now we have a clever way of maximizing a MI term, but it is indeed doing something very similar to HER from a technical contribution?
> > >
> > > Further comment : I don't necessarily agree, or fully understand this from a technical perspective - "We instead would like for this policy to be reusable, by, e.g., being able to control what state it reaches".
> > >
> > > What do you exactly mean by a policy to be reusable? Is it more like a transfer learning setting? If so, are there any experiments that justify that using the trained policy in a new task is useful?
> > >
> > > Overall, as you mention "first uniformly set goals over the state space (maximize H(g)) and then separately learn to reach those goals (minimize H(g | s))" - this seems to me like a very minor modification to several other tons of papers related to HER - where we randomly sample goal states and encourage the agent to reach these goal states. This paper seems like a very marginal contribution compared to that.

---

> > > > ### Author Response · Authors · 2019-11-13
> > > > **Re: Clarifications, further comments... from AnonReviewer1 (1/2)**
> > > >
> > > > Thank you for the additional response. We answer your remaining questions, and are happy to continue discussing if there are still points of confusion. In particular, we explain the large empirical and conceptual differences between Skew-Fit and prior methods, both of which we believe would be of interest to the ICLR community.
> > > >
> > > >
> > > > > How does Skew-Fit relate to hindsight experience replay (HER) and similar approaches? How is Skew-Fit different?
> > > > We note that there are significant empirical differences between Skew-Fit and HER. In Figure 5, we see that HER does not perform well without access to an oracle uniform goal distribution (more on this below). Specifically, HER has a final distance that is 400%, 200%, and 150% higher than the final distance when using of Skew-Fit, on the door, pickup, and pushing tasks, respectively. Similarly, Skew-Fit outperforms other prior methods across all tasks.
> > > >
> > > > A major conceptual difference between Skew-Fit and many prior methods, such as HER, is that we *learn* the goal distribution, whereas many prior methods *assume* that a uniform goal distribution is provided. When using images as observations, this amounts to assuming that the agent knows the distribution of natural images -- an unreasonable assumption in most cases. For example, in the object-pushing task of the hindsight experience replay (HER) paper, an XY goal-position for the object is sampled uniformly from within the workspace of the robot and given to the robot for exploration. While this procedure is simple to implement in a simulated domain and when the goal corresponds to an XY-Cartesian position in the plane, it is much more challenging when goals correspond to images, as in the environments that we tested. Randomly sampling an image will result in an image of static noise. So, Skew-Fit learns a goal distribution that corresponds to the uniform distribution over the set of valid states, instead of assuming that we are given access to this goal distribution.
> > > >
> > > > Because most prior goal-conditioned RL methods assume access to an oracle uniform goal distribution, most of the methods have only been applied to simulated domains, where defining such a goal distribution by hand is easy. As our experiments show, without access to a uniform distribution, prior methods such as HER perform poorly. Moreover, many of these methods (AutoGoal GAN, Rank-Based Prioritization, HER) assume access to ground-truth state information for computing the reward, which is readily available only in simulation. Enabling goal-conditioned RL to be applied to domains where the goal-space and reward function are unknown a priori, such as image-based domains, is important if we want to use these methods outside of simulation and for real-world applications. We believe that Skew-Fit is a useful step, both empirically and conceptually, towards this objective. Note that we evaluate Skew-Fit on a real-world image-based robotic manipulation task to demonstrate this (see Figure 1 and 7).
> > > >
> > > > As far as we know, the only other goal-conditioned methods that have been developed for goal images are RIG and DISCERN, but neither of these methods address the important question of how goals should be sampled for exploration. Applying Skew-Fit results in considerable performance gains over these prior methods: Figure 5 shows that RIG and DISCERN have final distances that are about 100% higher than that of Skew-Fit. Lastly, the only method that has been applied to real-world robot domains from images is RIG, and we found that Skew-Fit outperformed it not only in simulation, but also on the real-world door task.
> > > >
> > > >
> > > > > How are these goals determined in the first place?
> > > > We first describe how we generate goals chosen for evaluation. For the simulated tasks, we use an “oracle” sampling procedure that exploits the fact that the task is simulated. Note that this procedure is only used for evaluation and never used by the algorithms. To generate an evaluation goal, we sample a ground-truth state uniformly distribution from the entire state space, set the environment to this state, capture an image corresponding to this state, and use the resulting image as the goal. We then return the environment state back to its original state and instruct the policy to reach the captured goal image. This procedure based on ground-truth state information is only used for evaluation and in simulation. For the real-world door task, we took images of the door at 5 different angles, evenly spaced from 0 to 45 degrees. Like before, these goals are only used for evaluation and not be the algorithms.
> > > >
> > > > We now describe how goals for exploration are generated. At the very beginning of exploration, the agent takes random actions in the environment to collect a set of states. From thereon, the exploration goals are generated by the agent by sampling from the learned goal distribution, which is learned with Skew-Fit.
> > > >
> > > > (continued in next comment)

---

> > > > > ### Author Response · Authors · 2019-11-13
> > > > > **Re: Clarifications, further comments... from AnonReviewer1 (2/2)**
> > > > >
> > > > > (continued from previous comment)
> > > > >
> > > > > > What do you exactly mean by a policy to be reusable?
> > > > > By reusable, we mean that a policy can reach user-defined goals after performing exploration. In other words, it is a goal-conditioned policy rather than a policy that only performs exploration. For example, imagine you left a robot in a large office. If it performs exploration well, then the robot will autonomously visit every room in the building, regardless of whether it is goal-oriented or not. The difference between a goal-conditioned and non-goal-conditioned exploration policy is based on what happens next: After exploration, you can tell a goal-conditioned policy, “Please go to room A.” and it will know how to go to room A since it has already practiced reaching every possible goal. In other words, we can *reuse* the exploration policy to achieve these user-specified goals. However, if the exploration policy is not goal-conditioned but instead trained with an exploration-reward bonuses, then there is no way to control the policy. It may have visited every room during exploration, but at any given time, it only knows how to reach one location. In particular, it will always go to the last state that was deemed novel by the exploration-reward bonus.
> > > > >
> > > > >
> > > > > > What do you exactly mean by a distribution over terminal states? Why not the normalized discounted weighting of states?
> > > > > By terminal state, we mean the last state of each episode. So, the distribution over terminal states is the distribution of states where the policy will be located at the end of each episode. For example, will the policy always end at position X? Or will its final position have a Gaussian distribution? Uniform distribution?
> > > > >
> > > > > For our analysis, we found it more natural to study the distribution over terminal states, since in the goal-conditioned setting, we would like our goal-conditioned policy to *end* at a goal state when the episode is complete. However, we believe that our analysis could be applied to normalized discounted weighting of states by allowing an agent to constantly set new goals rather than waiting until the beginning of a new episode to set the next goal.
> > > > >
> > > > >
> > > > > > Does “no user-specified rewards” mean a sparse reward?
> > > > > No, and we understand that this may have been a source of confusion. By “no user-specified rewards” we mean that the user does not need to manually engineer a reward for each task. Instead, as described in Section 4, we use the same reward as the one used in RIG, which is an approximation of the log probability of the goal given the current state. Overall, this means that the same code and generic reward is used for the real-world door task and all of the simulated tasks. For example, there is no reward that specifically tells the robot to open the door, nor is there a reward that specifically tells the robot to pick up or move the objects. Instead, the same Skew-Fit objective encourages the robot to learn to manipulate these objects into as many configurations as possible (by setting diverse goals and then reaching those goals) regardless of the environment.

---

> > > > > > ### Author Response · Authors · 2019-11-15
> > > > > > **Additional Experiments Added**
> > > > > >
> > > > > > As suggested, we have also added additional exploration experiments on a simple maze setup in Section B.1. The maze and action spaces are designed so that random actions are unlikely to result in fast exploration and instead require goal-directed exploration. In these new experiments, we see that Skew-Fit significantly accelerates exploration.
> > > > > >
> > > > > > We have also added a simulated robot quadruped experiment that requires a robot to explore a narrow box-shaped corridor. In this experiment, we against see that Skew-Fit results in faster exploration than prior methods.
> > > > > >
> > > > > > We note that prior work in the field have similarly tested their algorithms on 3 simulated domains [1,2]. We believe that with our real-world robot experiments, as well as the additional experiments described above, our evaluation provides a similar level of rigor.
> > > > > >
> > > > > > [1] Andrychowicz, Marcin, et al. "Hindsight experience replay." Neural Information Processing Systems. 2017.
> > > > > > [2] Nair, Ashvin, et al. “Visual Reinforcement Learning with Imagined Goals. Neural Information Processing Systems. 2019.

---

### Official Review · AnonReviewer4 · 2019-10-21
**Official Blind Review #4**

**Rating:** 6

**Review:**

The paper introduces SKEW-FIT, an exploration approach that maximizes the entropy of a distribution of goals such that the agent maximizes state coverage.

The paper is well-written and provides an interesting combination of reinforcement learning with imagined goals (RIG) and entropy maximization. The approach is well motivated and simulations are performed on several simulated and real robotics tasks.

Some elements were unclear to me:
- "We also assume that the entropy of the resulting state distribution H(p(S | pφ)) is no less than the entropy of the goal distribution H(pφ(S)). Without this assumption, a policy could ignore the goal and stay in a single state, no matter how diverse and realistic the goals are." How do you ensure this in practice?
- In the second paragraph of 2.2, it is written "Note that this assumption does not require that the entropy of p(S | pφ) is strictly larger than the entropy of the goal distribution, pφ." Could you please clarify?


The experiments are interesting, yet some interpretations might be too strong (see below):
- In the first experiment, "Does Skew-Fit Maximize Entropy?", it is empirically illustrated that the method does result in a high-entropy state exploration. However, it is only compared to one very naive way of exploring and it is not discussed whether other techniques also achieve the same entropy maximization. The last sentences seems to imply that only this technique ends up optimizing the entropy of the state coverage, while I believe that the claim (given the experiment) should only be about the fact it does so faster.
- On the comments of Figure 6, the paper mentions that "The other methods only rely on the randomness of the initial policy to occasionally pick up the object, resulting in a near-constant rate of object lifts." I'm unsure about the interpretation of this sentence given Figure 6 because other methods do not seem to fail entirely when given enough time.
- In the experiment "Real-World Vision-Based Robotic Manipulation", It is written that "a near-perfect success rate [is reached] after five and a half hours of interaction time", while on the plot it is written 60% cumulative success after 5.5 hours and it is thus not clear where this "5.5 hours" comes from.

**Experience Assessment:**

I have published in this field for several years.

**Review Assessment: Checking Correctness Of Derivations And Theory:**

I assessed the sensibility of the derivations and theory.

**Review Assessment: Checking Correctness Of Experiments:**

I assessed the sensibility of the experiments.

**Review Assessment: Thoroughness In Paper Reading:**

I read the paper thoroughly.

---

> ### Author Response · Authors · 2019-11-07
> **Re:  Official Blind Review #4**
>
> Thank you for the review and suggestions. We have adjusted the experimental discussion to clarify a few points of confusion and to avoid possibly overstating the results.
>
>
> > "We also assume that the entropy of the resulting state distribution H(p(S | pφ)) is no less than the entropy of the goal distribution H(pφ(S)). Without this assumption, a policy could ignore the goal and stay in a single state, no matter how diverse and realistic the goals are." How do you ensure this in practice?
> We found that using RIG performed quite well. In particular, we found that using hindsight experience replay with the dense latent-distance ensured that the goal-conditioned policies consistently paid attention to the goal, and eventually learned to reach them.
>
>
> > In the second paragraph of 2.2, it is written "Note that this assumption does not require that the entropy of p(S | pφ) is strictly larger than the entropy of the goal distribution, pφ." Could you please clarify?
> We mean that the entropy of p(S | pφ) and p(φ) can be equal. It is unnecessary for the entropy to increase during exploration, since we increase it by changing the goal-distribution.
>
>
> > The last sentences seems to imply that only this technique ends up optimizing the entropy of the state coverage, while I believe that the claim (given the experiment) should only be about the fact it does so faster.
> We agree that other methods can also eventually maximize the entropy. We have modified the sentence to clarify that we mean that Skew-Fit results in higher entropy faster.
>
>
> > I'm unsure about the interpretation of this sentence given Figure 6 because other methods do not seem to fail entirely when given enough time.
> Thank you for pointing out this unclear phrasing. Since we are plotting the cumulative pickups, the success rate is given by the slope of the curves. While some prior methods do perform better than others, most curves have constant slopes after the first 40k iterations, meaning that their success rate does not increase over time. We have modified the text to clarify this.
>
>
> > it is thus not clear where this "5.5 hours" comes from.
> We have corrected the text to say 6 hours. Thank you!

---

### Official Review · AnonReviewer2 · 2019-10-22
**Official Blind Review #2**

**Rating:** 6

**Review:**

This paper introduced a very interesting idea to facilitate exploration in goal-conditioned reinforcement learning. The key idea is to learn a generative model of goal distribution to match the weighted empirical distribution, where the rare states receive larger weights. This encourages the model to generate more diverse and novel goals for goal-conditioned RL policies to reach.

Pros:
The Skew-Fit exploration technique is independent of the goal-conditioned reinforcement learning algorithm and can be plugged in with any goal-conditioned methods. The experiments offer a comparison to several prior exploration techniques and demonstrate a clear advantage of the proposed Skew-Fit method. It is evaluated in a variety of continuous control tasks in simulation and a door opening task on a real robot. A formal analysis of the algorithm is provided under certain assumptions.

Cons:
The weakest part of this work is the task setup. The method has only been evaluated on simplistic short-horizon control tasks. It’d be interesting to see how this method is applied to longer-horizon multi-stage control tasks, where exploration is a more severe challenge. It is especially when the agent has no access to task reward and only explores the environment to maximize state coverage. It is unclear to me how many constraints are enforced in the task design in order for the robot to actually complete the full tasks through such exploration.

I would also like to see how Skew-Fit works with different goal-conditioned RL algorithms, and how the performances of the RL policy in reaching the goals would affect the effectiveness of this method in exploring a larger set of states.

Section E: it seems that there’s a logic jump before the conclusion “goal-conditioned RL methods effectively minimize H(G|S)”. More elaboration on this point is necessary.

Minor:
Appendix has several broken references.

**Experience Assessment:**

I have published one or two papers in this area.

**Review Assessment: Checking Correctness Of Derivations And Theory:**

I assessed the sensibility of the derivations and theory.

**Review Assessment: Checking Correctness Of Experiments:**

I assessed the sensibility of the experiments.

**Review Assessment: Thoroughness In Paper Reading:**

I read the paper at least twice and used my best judgement in assessing the paper.

---

> ### Author Response · Authors · 2019-11-07
> **Re: Official Blind Review #2**
>
> Thank you for the review and suggestions. Below, we address a number of questions asked and are happy to continue the discussion.
>
> > I would also like to see how Skew-Fit works with different goal-conditioned RL algorithms
>
> We are currently running experiments that replace SAC with TD3. The only image-based, goal-conditioned RL algorithm other than RIG that we are aware of is DISCERN, which we found never learned. We are happy to take suggestions for alternate image-based, goal-conditioned RLs algorithm to try.
>
>
> > More elaboration on this point is necessary.
>
> Thank you for the suggestion. We have updated Section E to clarify, and include the new text here for convenience:
> “...one can see that goal-conditioned RL generally minimizes H(G | S) by noting that the optimal goal-conditioned policy will deterministically reach the goal. The corresponding conditional entropy of the goal given the state, H(G | S) would be zero, since given the current state, there would be no uncertainty over the goal (the goal must have been the current state since the policy is optimal). So, the objective of goal-conditioned RL can be interpreted as finding a policy such that H(G | S) = 0. Since zero is the minimum value of H(G | S), then goal-conditioned RL can be interpreted as minimizing H(G | S).”
>
>
> > Appendix has several broken references.
>
> Thank you. We have fixed the references.

---

> > ### Author Response · Authors · 2019-11-15
> > **Additional Experiments Added**
> >
> > As suggested, we have added experiments that study the importance of the reinforcement learning algorithm used with Skew-Fit. Specifically, we replaced soft actor critic (SAC) with twin delayed deep deterministic policy gradient (TD3) and reran the simulated, vision-based experiments with this new combination. The results show that Skew-Fit performs well with both TD3 and SAC, with both versions achieving approximately the same final error. These results suggest that the benefits of Skew-Fit are not specific to SAC, and can instead be combined with other reinforcement learning algorithms.
> >
> > We have also added an experiment that explicit tests for exploration, by using a maze environment with long corridors and measuring the state coverage. These experiments are shown in Section B.1, and show that Skew-Fit significantly accelerates exploration. We have also added a simulated robot quadruped experiment that requires a robot to explore a narrow box-shaped corridor. In this experiment, we again see that Skew-Fit results in faster exploration than prior methods (see Section B.1).

---

### Decision · Program_Chairs · 2019-12-19

**Decision:**

Reject

**Comment:**

This paper tackles the problem of exploration in RL. In order to maximize coverage of the state space, the authors introduce an approach where the agent attempts to reach some self-set goals. The empirically show that agents using this method uniformly visit all valid states under certain conditions. They also show that these agents are able to learn behaviours without providing a manually-defined reward function.

The drawback of this work is the combined lack of theoretical justification and limited (marginal) algorithmic novelty given other existing goal-directed techniques. Although they highlight the performance of the proposed approach, the current experiments do not convey a good enough understanding of why this approach works where other existing goal-directed techniques do not, which would be expected from a purely empirical paper. This dampers the contribution, hence I recommend to reject this paper.